# Hierarchies over Pixels: A Benchmark for Cognitive Geospatial Reasoning for Agents

## Abstract

Beyond perception, reasoning is crucial in remote sensing, enabling advanced interpretation, inference, and decision-making. Recent advances in large language models (LLMs) have given rise to tool-augmented agents that enhance reasoning by leveraging external tools for complex analytical tasks. However, existing research on these agents in remote sensing largely focuses on perception-oriented tasks, with cognitive geospatial reasoning remaining underexplored. In this work, we systematically evaluate the geospatial reasoning capabilities of LLM-powered tool-augmented agents. To this end, we introduce **GeoHOP**, a benchmark for hierarchical geospatial reasoning. GeoHOP comprises 545 scenario-driven, hierarchy-aware tasks—such as hazard vulnerability assessment, urban heat island analysis, and forest fragmentation dynamics—spanning optical, Synthetic Aperture Radar (SAR), and infrared (IR) imagery. GeoHOP advances evaluation beyond monitoring-based recognition to cognitive-level geospatial analysis. Building upon GeoHOP, we propose **GeoPlanner**, an agent powered by LLMs that organizes 5 toolkits into functional hierarchies and executes fault-tolerant reasoning pipelines. GeoPlanner enables structured abstraction, robust recovery from tool failures, and stable long-horizon planning. Extensive experiments across diverse geospatial reasoning tasks demonstrate that GeoPlanner excels in hierarchical reasoning, cross-modal transfer, and error handling.

## 1 Introduction

Large language models (LLMs), empowered by generative pretraining and instruction tuning, have substantially advanced zero-shot task completion across diverse applications (Yang et al., 2024; Zhou et al., 2024). Building on this progress, LLM-driven agents can decompose goals into sub-tasks and orchestrate external tools, enabling robust multi-step workflows (Zhao et al., 2024; Li, 2025). However, as tasks demand increasingly granular understanding—particularly in remote sensing (RS) scenarios—these general-domain agents encounter substantial limitations. Their performance degradation stems from RS-specific challenges, including heterogeneity across modalities (optical, Synthetic Aperture Radar (SAR), infrared) and extensive variation in object size, scale, and orientation across diverse landscapes worldwide.

To address these challenges, researchers have begun adapting LLMs to RS through tool-augmented agents. Examples include Remote Sensing ChatGPT(Guo et al., 2024), RS-Agent(Xu et al., 2024), GeoLLM-Engine(Singh et al., 2024), Change-Agent(Liu et al., 2024a), Tree-GPT(Du et al., 2023). While effective for perception-oriented tasks like classification, localization, counting, and visual question answering, these approaches remain confined to visual perception. They fall short of addressing the cognitively demanding reasoning needed in realistic geospatial applications.

Beyond perception, reasoning is crucial for informed decision-making and advanced scene interpretation in Earth observation. For instance, geospatial reasoning enables identifying buildings in proximity to water bodies for flood-risk screening (Oubennaceur et al., 2019), quantifying the proportion of cropland adjacent to water sources for irrigation assessment (Fu et al., 2022), or detecting barren patches that are fully enclosed by forest to identify internal clearings requiring ecological stabilization (Hansen et al., 2009). These scenarios underscore the need for reasoning capabilities that extend beyond perception, requiring models to reason over complex spatial relationships and execute multi-step analytical workflows.

Despite recent efforts, systematic evaluation of geospatial reasoning in LLM-driven agents remains limited. For instance, ThinkGeo(Shabbir et al., 2025) introduces a benchmark for geospatial reasoning, but its scope is confined to optical imagery, leaving multi-modal reasoning underexplored. To bridge this gap, we introduce GeoHOP, a benchmark explicitly designed for cognitive-level geospatial reasoning across multiple modalities. GeoHOP consists of 545 scenario-driven, hierarchy-aware tasks spanning multiple modalities—including optical, SAR, and infrared imagery—and encompassing complex planning scenarios such as hazard risk estimation, urban heat island analysis, and ecosystem fragmentation detection. This benchmark enables rigorous evaluation across perception, geospatial reasoning, and advanced decision-making.

Building on **GeoHOP**, we further propose **GeoPlanner**, an LLM-driven agent that organizes analytic tools into functional hierarchies and executes fault-tolerant reasoning pipelines. GeoPlanner facilitates structured abstraction, dynamic workflow tracking, and robust recovery from intermediate failures, enabling cognitive reasoning in complex geospatial tasks.

Our main contributions are summarized as follows:

- We present **GeoHOP**, the first benchmark for geospatial reasoning, comprising 545 tasks across optical, SAR, and infrared modalities.

- We propose **GeoPlanner**, an agentic framework that enables cognitive reasoning with hierarchical planning, robust error handling, and support for multi-modal RS scenarios.

- We establish new baselines by comprehensively evaluating state-of-the-art LLMs on **GeoHOP**, revealing both their strengths and limitations in cognitively demanding geospatial reasoning.

## 2 RELATED WORK

The extension of LLMs to RS has attracted growing interest, giving rise to a range of approaches encompassing interactive assistants, domain-specific frameworks, modular toolchains, and foundation-model paradigms. Early works primarily target specialized tasks: for example, TreeGPT (Du et al., 2023) addresses forestry applications via individual tree segmentation and ecological parameter extraction, while Change-Agent (Liu et al., 2024a) supports change detection and captioning, enabling interactive interpretation of changed regions.

Recent efforts aim to create more general-purpose RS agents. Remote Sensing ChatGPT (Guo et al., 2024) integrates ChatGPT (Brown et al., 2020) with pretrained RS networks to handle a range of perception-oriented tasks. RS-Agent (Xu et al., 2024) expands the task spectrum via scalable tool integration, handling workflows that require specialized domain expertise. In parallel, GeoLLM-Engine (Singh et al., 2024) leverages fully operational APIs with dynamic map and web interfaces to execute geospatial tasks, while UnivEARTH (Kao et al., 2025) curates domain-grounded QA tasks from NASA Earth Observatory articles to evaluate the ability of LLMs to generate executable Earth Engine code.

Beyond RS, a broad spectrum of reasoning benchmarks has been developed to evaluate LLM capabilities.FOLIO (Han et al., 2024) evaluates natural language reasoning mapped to first-order logic (FOL), with expert-written premises and conclusions paired with syntactically validated FOL formulas. It emphasizes complex logical patterns and multi-step deduction chains. LogiCity (Li et al., 2024a) offers a neuro-symbolic urban simulation governed by customizable FOL rules, with tasks spanning long-horizon sequential decision-making and single-step visual reasoning under perceptual noise. Both benchmarks assess an LLM's ability to learn abstractions, generalize to new entity compositions, and reason from symbolic predicates — skills relevant to RS but applied in non-geospatial contexts. Compared to these benchmarks, geospatial reasoning imposes additional complexities: (i) Multi-modal perception — heterogeneous sensing modalities (optical, SAR, IR) with varying resolutions and noise characteristics; (ii) Spatially grounded logic — reasoning tied to geographic coordinates, projections, topology, and immutable physical constraints; (iii) Specialized geospatial operations — workflows requiring chaining of domain-specific operators (e.g., SAR classification + buffer + overlap) under legality constraints; (iv) Multi-scale context — phenomena spanning from local object detection to continental-scale environmental change over time. These distinctions mean that while LLMs excelling in FOLIO or LogiCity may demonstrate strong general logical reasoning, they can still struggle to adapt when perception involves domain-unique data types and reasoning

depends on spatially explicit, physically grounded models — underscoring the need for RS-specific benchmarks.

Despite these advances, existing frameworks often lack planning transparency and fine-grained step-level reasoning. To date, ThinkGeo (Shabbir et al., 2025) is the only work that introduces step-wise evaluation protocols for perception, planning, and geospatial reasoning. However, its coverage is restricted to optical imagery, with no support for additional modalities such as SAR and limited handling of specialized geospatial operations.

## 3 GEOHOP DATASET

We propose **GeoHOP**, a benchmark designed to assess the geospatial reasoning capabilities of tool-augmented agents powered by LLMs.GeoHOP integrates diverse imagery from optical, SAR, and infrared modalities with expert-curated knowledge and tool-augmented query pipelines, yielding 545 high-quality instances.

Each instance couples real-world imagery with structured, multi-step reasoning challenges that require both low-level perception (e.g., segmentation, detection) and high-level decision-making (e.g., urban planning, disaster assessment). Unlike prior remote sensing benchmarks that focus narrowly on perception (e.g., classification or detection), GeoHOP emphasizes *end-to-end reasoning*, from perception through spatial analysis to actionable conclusions, thereby providing a rigorous testbed for agents. Figure 1 illustrates six representative samples from the GeoHOP benchmark.

### 3.1 GEOSPATIAL REASONING SCENARIOS

A central challenge in RS is bridging the gap between low-level perceptual outputs and the high-level intelligence required for real-world decision-making. To address this, we adopt a structured taxonomy of geospatial reasoning tasks, systematically organized into seven primary domains: urban planning, disaster assessment, environmental monitoring, transportation analysis, activation monitoring, maritime monitoring, and industrial sites.

### 3.2 DATA CONSTRUCTION PIPELINE

We construct GeoHOP with a two-stage pipeline (see Figure 2) that combines diverse source imagery, knowledge-augmented scenario generation, and multi-pass expert adjudication to produce 545 validated instances. The source datasets used in this pipeline are summarized in Table 1.

**Stage 1: Knowledge-augmented scenario generation.** We obtain a stratified sample of source imagery (by modality and scene type) and apply explicit hardness controls to select cases that require multi-tool reasoning and compositional spatial relations (e.g., proximity, containment, topology). Candidate queries and tool-chains are generated with ChatGPT-5 via in-context learning: prompts are modality-specific and seeded with expert-authored exemplars.

To ground generation in domain knowledge, we inject a compact knowledge corpus into prompts. The corpus covers four guidance categories: (i) urban greening and heat-mitigation frameworks (Twohig-Bennett & Jones, 2018; Bowler et al., 2010; Aram et al., 2019; Norton et al., 2015; Rigolon, 2016); (ii) international land-cover standards (Di Gregorio & Jansen, 1998; Mosca et al., 2020); (iii) aviation and maritime search-and-rescue doctrine for IR small-target tasks (Kim et al., 2020); and (iv) industrial safety rules for separation and proximity of hazardous assets (Ricci et al., 2021; Kuk-fisz et al., 2022). These anchors force the ChatGPT-5 to produce quantitative thresholds (distances, areas, class compositions) consistent with established frameworks. We also enforce operator legality checks (including ground distance sampling (GSD)-aware units) to prevent invalid tool sequences. See Appendix A.6 SCENE-CONTEXT KNOWLEDGE BASE for the details.

**Stage 2: Expert adjudication and refinement.** All generated candidates undergo a hierarchical, three-pass review by a panel of eight remote-sensing experts. Reviewers apply an auditable rubric that scores (1) *semantic integrity* — whether the query is meaningful and relevant; (2) *tool-sequence coherence* — whether the proposed tool-chain and parameters are logically consistent and GSD-

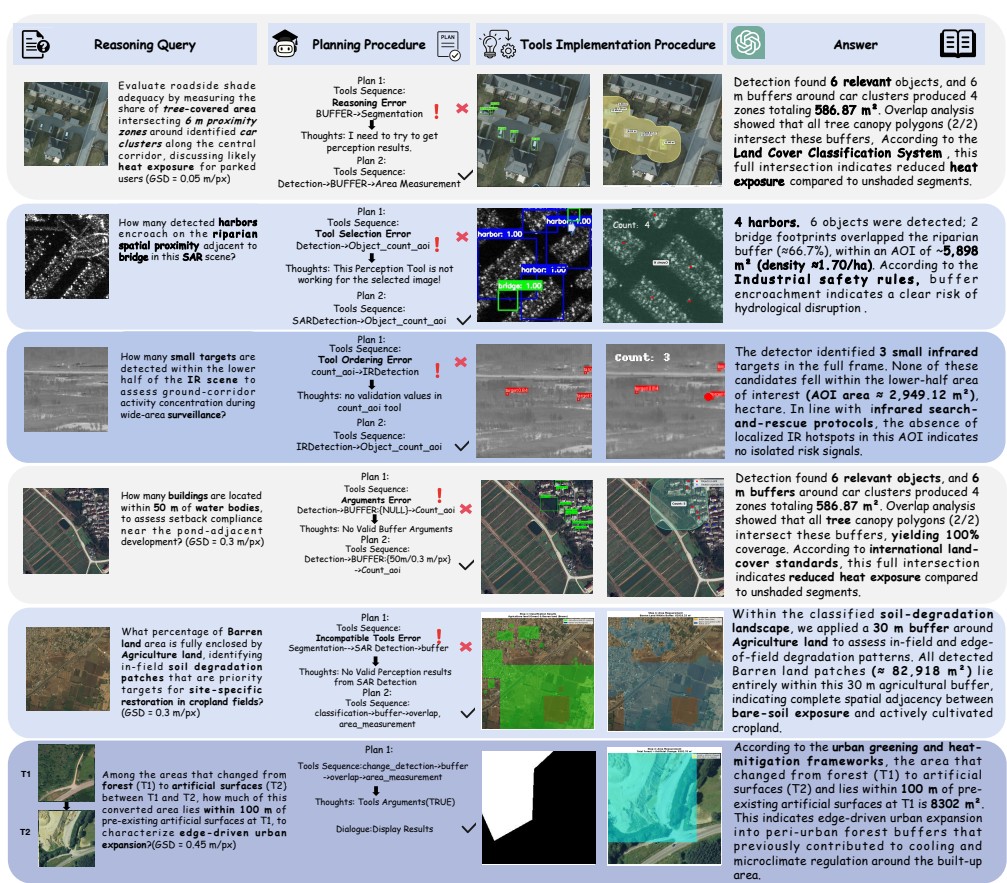

Figure 1: Representative GeoHOP tasks and GeoPlanner execution traces under our error taxonomy. From left to right: (i) input image and query, (ii) high-level plans from the Geo planner, (iii) the executed tool sequence, and (iv) the final geospatial answer from the tool outputs. Red crosses mark erroneous intermediate plans or tool calls (e.g., reasoning, selection, ordering, argument, or compatibility errors), while green checkmarks indicate the validated final tool chain and answer.

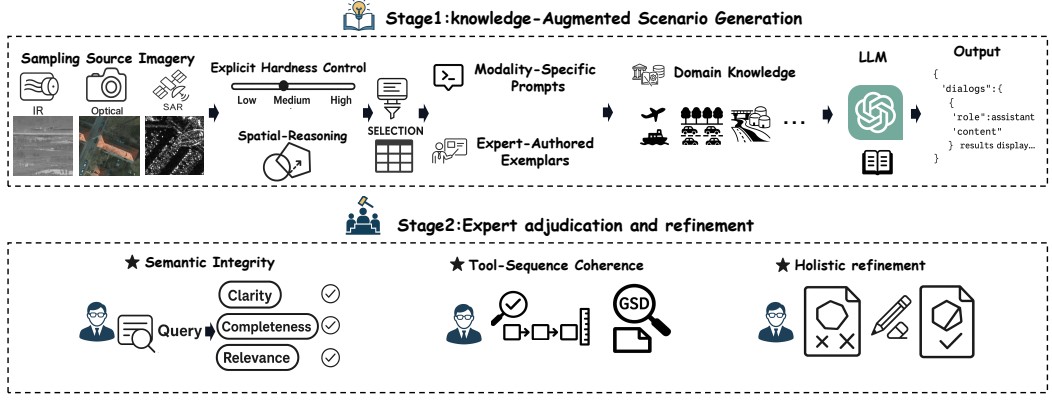

Figure 2: Pipeline for constructing the GeoHOP benchmark.

aware; and (3) *holistic refinement* — correction of ambiguous phrasing, geometry errors, or minor inconsistencies. Every edit is recorded in a curation interface to preserve an audit trail.

| Name | Annotation Type | Resolution | Modality | Geographical Coverage |
|------|-----------------|------------|----------|----------------------|
| LoveDA (Wang et al., 2021) | Masks | 0.3 m/px | Optical | Nanjing/Changzhou/Wuhan, China |
| ISPRS Potsdam (Song & Kim, 2020) | Masks | 0.05 m/px | Optical | Potsdam, Germany |
| ISPRS Vaihingen (isp, 2014) | Masks | 0.09 m/px | Optical | Vaihingen, Germany |
| DeepGlobe Land Cover (Li & Chen, 2019) | Masks | 0.50 m/px | Optical | Cities in India/Indonesia/Thailand |
| HRSCD (Daudt et al., 2018) | Change masks | 0.45 m/px | Optical | Rennes/Caen, France |
| OGSOD (Wang et al., 2023) | Bounding boxes | 3 m/px | SAR | Cities in China |
| DMIST (Chen et al., 2024) | Bounding boxes | – | IR | Chengdu, China |

Table 1: Datasets used as image sources in the GeoHOP benchmark.

| Metric | All | Optical | SAR | IR |
|--------|-----|---------|-----|-----|
| Total queries | 545 | 386 | 116 | 43 |
| Total tool calls | 1649 | 1217 | 346 | 86 |
| 2/3/4-tools use | 72/387/86 | 27/273/86 | 2/114/0 | 43/0/0 |
| Maximum question length | – | 406 | 207 | 288 |
| Minimum question length | – | 109 | 111 | 135 |
| Average question length | 158.24 | 160.13 | 143.49 | 181.05 |

Table 2: Statistics of question complexity and tool-use patterns across different modalities.

## 3.3 BENCHMARK STATISTICS

GeoHOP contains 545 queries across optical, SAR, and IR modalities. In total, 13 tools are invoked 1649 times, with most queries requiring multi-tool composition. Query lengths range from short factual questions to complex compositional ones, demonstrating both diversity and reasoning depth. Detailed statistics are reported in Table 2.

Each reasoning task in GeoHOP is instantiated as a distinct scene context type, characterized by three components: (i) a SPATIAL TRIGGER (quantitative thresholds or topological relations), (ii) an EXPERT INTERPRETATION (domain-grounded semantics), and (iii) a VERIFIABLE TOOL-CHAIN. This hierarchical design bridges perception and cognition, offering structured priors and explicit multi-step pathways while ensuring interpretability and verifiability. By covering both canonical and high-complexity reasoning scenarios, GeoHOP provides rigorous testbeds for evaluating diverse agentic capabilities, including fine-grained spatial understanding, multi-step tool composition, quantitative analysis, and context-aware risk assessment.

## 4 THE GEOPLANNER AGENT

We propose **GeoPlanner**, an agent tailored for cognitively demanding geospatial reasoning in RS. GeoPlanner's LLM controller orchestrates an end-to-end workflow (see Figure 3): (i) semantic retrieval of a task-relevant toolset from a hierarchical, typed tool library; (ii) abstract, modality-aware plan generation over the retrieved tools; (iii) execution-time operator selection and parameterization with error-aware adaptation (within-toolkit substitution and prefix-preserving replanning); and (iv) verifiable answer synthesis strictly from structured tool outputs.

GeoPlanner extends the toolkit paradigm (Liu et al., 2024b) and agent framework (Fallahpour et al., 2025) to geospatial analysis by replacing flat tool sets with a multi-level, domain-specific hierarchy that incorporates typed I/O and geospatial constraints, thereby enabling reliable, modality-aware reasoning across optical, SAR, and IR data. GeoPlanner is explicitly designed to meet four RS-specific requirements: (1) *modality awareness* enforced in both retrieval and composition; (2) *hierarchical abstraction* with typed operators plus spatial unit checks and legality constraints; (3) *fault tolerance* via within-toolkit substitution and prefix-preserving replanning with structured error context; and (4) *verifiable grounding*, validating spatial outputs (e.g., geometry validity), computing all quantitative results with tools, and having the LLM synthesize interpretable, context-aware analyses by integrating validated tool I/O with pre-encoded scenario knowledge.

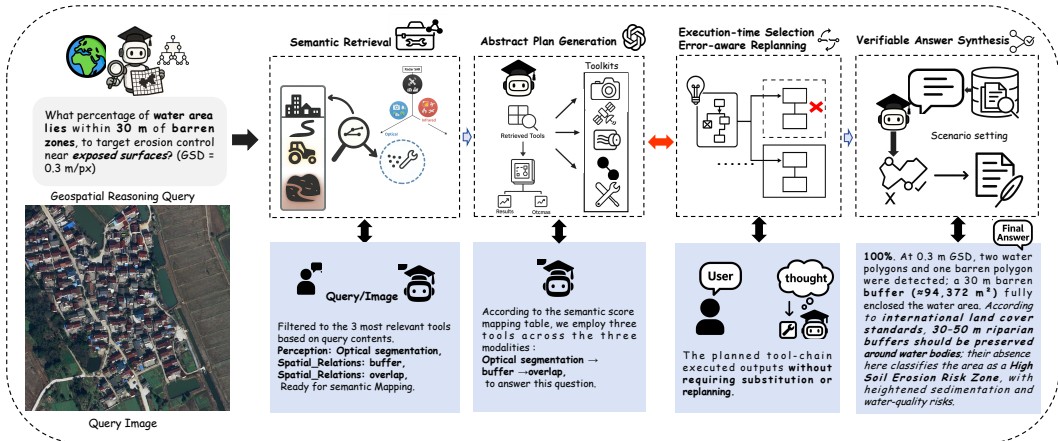

Figure 3: End-to-end workflow of the GeoPlanner Agent.

## 4.1 SEMANTIC RETRIEVAL OVER A HIERARCHICAL, TYPED TOOL LIBRARY

We organize all tools into a multi-level hierarchy $\mathcal{H}$ with **five** top-level toolkits: (1) *Perception* (Optical segmentation, Optical detection, Optical classification, Optical change detection ), (2) *Spatial-Relations* (buffer, overlap, containment), (3) *Spatial-Statistics* (distance calculation, area measurement, object counting), (4) *SAR Tools* (SAR detection, SAR classification), and (5) *IR Tools* (IR detection). Each operator advertises (i) typed I/O (e.g., vector geometries, rasters, masks, bounding boxes), (ii) spatial unit requirements, and (iii) legality constraints, and is annotated with modality tags.

Given a natural-language query $Q$, a semantic retriever $\mathcal{R}$ aligns $Q$ with textual tool descriptors in $\mathcal{H}$ using sentence-transformer embeddings (Reimers & Gurevych, 2019) to induce a candidate set:

$$\mathcal{C}_{\text{candidate}} = \mathcal{R}(Q, \mathcal{H}).$$

This step reduces the planning search space, enforces modality consistency, and prevents invalid compositions at the outset.

**Integration of external expert models.** GeoPlanner grounds high-level plans in verifiable computations by integrating state-of-the-art open-source models into the hierarchy: **RemoteSAM** (Yao et al., 2025) (*Perception*) for unified optical segmentation/detection/classification over 297 fine-grained categories; **SARATR-X** (Li et al., 2025; 2024b) (*SAR Tools*), a foundation model pretrained on 180K SAR samples for robust detection and classification; **DMIST/LASNet** (Chen et al., 2024) (*IR Tools*) for infrared target detection; and **Change3D** (Zhu et al., 2025) (*Change Detection*), a video-modeling–based framework that unifies binary, semantic, and damage-level change reasoning through learnable perception frames. Together with Spatial-Relations and Spatial-Statistics operators, these expert tools form GeoPlanner's operational backbone, ensuring reproducibility and scalability.

## 4.2 ABSTRACT PLAN GENERATION OVER THE RETRIEVED TOOLSET

Conditioned on $\text{desc}(\mathcal{C}_{\text{candidate}})$, the planner $\rho_\theta$ (few-shot prompted with task-decomposition instructions and a constrained action schema) produces a multi-step abstract plan

$$\mathcal{P} \sim \rho_\theta(Q, \text{desc}(\mathcal{C}_{\text{candidate}})).$$

Each step binds to an operator *family* in $\mathcal{C}_{\text{candidate}}$, declares expected I/O types and units, and defers parameterization (e.g., buffer distances, class labels, thresholds) to execution. This yields a modality-aware blueprint that preserves semantic validity while retaining flexibility for data-driven parameter selection.

### 4.3 EXECUTION-TIME SELECTION & ERROR-AWARE REPLANNING

During execution, the controller instantiates each abstract step by selecting a concrete operator $t$, binding arguments, and validating explicit success signals (e.g., non-empty geometries/masks and numeric stability). Failures are treated as informative signals and handled via two strategies:

**(1) Within-toolkit substitution.** GeoPlanner first selects a functionally similar alternative $t'$ inside the same toolkit to preserve the abstract workflow $\mathcal{P}$ with minimal disruption (e.g., switching from *overlap* to *containment* within Spatial-Relations).

**(2) Prefix-preserving replanning (error-aware).** If local substitution fails, the validated prefix of $\mathcal{P}$ is retained, while a structured error context $E$ (failed operator, arguments, logs, and validated intermediates) is injected into retrieval. A refreshed set $\mathcal{C}(Q, E)$ supports regeneration of the failing suffix. A step succeeds if any candidate succeeds ($\bigcup_{t \in \mathcal{C}(Q,E)} \text{succ}_t$); total failure occurs only when all fail ($\bigcap_{t \in \mathcal{C}(Q,E)} \text{fail}_t$). Retries are capped to bound cost, yielding a budget-conscious controller that preserves progress under adverse conditions.

### 4.4 VERIFIABLE ANSWER SYNTHESIS

GeoPlanner synthesizes the final answer strictly from validated tool outputs: (i) all numeric values (counts, areas, distances, proportions) are computed by operators with explicit spatial units; (ii) aggregations include provenance (modality, operator names) and spatial unit conversions when necessary; and (iii) the LLM only provides natural-language contextualization. This separation reduces hallucination risk and improves transparency and reproducibility.

## 5 EXPERIMENT

To evaluate the reasoning and tool-use capabilities of **GeoPlanner** under real-world RS scenarios, we conduct comprehensive evaluations on the **GeoHOP** benchmark. As the controller in GeoPlanner is powered by LLMs, We evaluate a diverse suite of foundation models, spanning both API-based systems—GPT-3.5 (Ouyang et al., 2022), GPT-4o (Hurst et al., 2024), and Gemini-2.5-Flash (Team et al., 2023)—and a broad range of open-source LLMs. The open-source group covers both large-capacity models (DeepSeek-V3 (Bi et al., 2024), Qwen2.5-32B-Instruct (Team, 2024)),LLaMA-3-70B-Instruct (Dubey et al., 2024) and medium-sized systems such as Qwen2.5-7B (Bai et al., 2023), InternLM3-8B (Cai et al., 2024), LLaMA-3-8B (Dubey et al., 2024), Mistral-7B (Jiang et al., 2023), Yi-1.5-6B (AI et al., 2025), and Phi-3-Mini (Abdin et al., 2024). All experiments are conducted on an NVIDIA A5000 GPU within the OpenCompass evaluation platform.

### 5.1 EVALUATION STRATEGY

We follow the ReAct-style (Yao et al., 2023) evaluation protocol, which includes both step-by-step and end-to-end modes. These protocols define how tool-augmented reasoning is assessed, and our evaluation criteria remain fully consistent with ReAct, ensuring comparability across different agents. In our framework, GeoPlanner orchestrates the workflow of tool invocations, which complements rather than contradicts the ReAct protocol by providing structured execution while adhering to its evaluation standards. Step-by-step metrics (InstAcc, ToolAcc, ArgAcc, and SummAcc) report the accuracy of instruction-following, tool selection, argument correctness, and summary generation, respectively. End-to-end scores (P, O, and L) reflect tool-augmented predictions within the *Perception*, *Spatial-Statistics*, and *Spatial-Relations* toolkits.

Unlike GTA (Wang et al., 2024), which computes final answer accuracy (AnsAcc) through deterministic string matching, we argue this approach is insufficient for GeoHOP's complex reasoning scenarios. String matching is brittle to minor lexical variations and may misclassify semantically correct but differently phrased predictions. This issue is particularly problematic in GeoPlanner, where final answers synthesize information from *query semantics, tool inputs/outputs, and scenario knowledge* rather than matching a single canonical string.

To address these limitations, we adopt **LLM-as-a-judge** for AnsAcc evaluation. For each query, we construct curated evaluation prompts and employ GPT-4o-mini as an automatic judge to assess

| Model | Step-by-Step Metrics (%) | | | | End-to-End Metrics (%) | | | |
|---|---|---|---|---|---|---|---|---|
| | Inst. | Tool. | Arg. | Summ. | P | O | L | Ans. |
| **API-based** | | | | | | | | |
| GPT-4o | **79.40** | 72.01 | **23.57** | **82.83** | **72.73** | **88.38** | 77.95 | **25.70** |
| Gemini-2.5-Flash | 59.09 | 43.31 | 15.75 | 47.22 | 41.47 | 40.89 | 41.51 | 12.04 |
| GPT-3.5 | 59.05 | 57.12 | 21.88 | 59.08 | 61.22 | 61.04 | 53.30 | 24.40 |
| **Open-source (Large)** | | | | | | | | |
| DeepSeek-V3 | 65.84 | 66.33 | 23.32 | 60.92 | 65.44 | 66.04 | 57.93 | 21.28 |
| Qwen2.5-32B-Instruct | 62.83 | 58.00 | 17.41 | 58.90 | 44.28 | 67.40 | 61.47 | 19.08 |
| Llama-3-70B-Instruct | 67.89 | 67.47 | 22.66 | 69.91 | 62.72 | 71.62 | 62.54 | 22.75 |
| **Open-source (Medium/Small)** | | | | | | | | |
| Qwen2.5-7B-Instruct | 77.19 | **74.71** | 23.11 | 65.32 | 68.56 | 73.03 | **82.42** | 22.57 |
| InternLM3-8B-Instruct | 61.26 | 57.72 | 22.59 | 50.28 | 61.44 | 53.70 | 60.70 | 18.90 |
| LLaMA3-1-8B | 69.46 | 65.99 | 22.27 | 55.60 | 68.42 | 60.34 | 70.82 | 16.70 |
| Phi-3-Mini-4K-Instruct | 55.45 | 41.14 | 17.17 | 10.09 | 43.63 | 7.52 | 51.74 | 3.67 |
| Mistral-7B-Instruct-v0.2 | 60.03 | 47.05 | 16.73 | 46.06 | 49.75 | 44.05 | 45.23 | 13.21 |
| Yi-1.5-6B-Chat | 63.38 | 42.08 | 16.00 | 37.98 | 47.44 | 38.73 | 36.02 | 10.83 |

Table 3: **Main results of GeoPlanner on the GeoHOP benchmark.** Inst., Tool., Arg., Summ. denote the step-by-step metrics InstAcc, ToolAcc, ArgAcc, and SummAcc respectively. P, O, and L denote the end-to-end scores of tool-augmented prediction in the *Perception*, *Spatial Statistics*, and *Spatial Relations* toolkits. **Ans.** denotes the final answer accuracy (AnsAcc). **Bold** indicates the best score among all models, and underline marks the best score within the same model scale.

whether predicted results and arguments are semantically consistent with ground truth. This approach provides a more reliable and context-aware measure of success in GeoHOP, aligning evaluation with the hierarchical, multi-fact reasoning required by real-world geospatial tasks while maintaining compatibility with the GTA evaluation framework.

## 5.2 MAIN RESULTS

We conduct experiments on the **GeoHOP benchmark** to comprehensively evaluate reasoning and tool-use abilities in real RS tasks. Unlike prior efforts relying on synthetic prompts or shallow tool interactions, GeoHOP introduces *multimodal, scenario-driven tasks* that compel agents to invoke multiple **toolkits** spanning perception, spatial analysis, SAR, and IR. These tasks are grounded in satellite and aerial imagery, requiring models to exhibit *hierarchical reasoning* and *quantitative precision* during multi-step execution, enabling systematic evaluation of cognitive-level geospatial reasoning.

Table 3 reports step-by-step (Inst., Tool., Arg., Summ.) and end-to-end (P, O, L, Ans.) performance on GeoHOP. Figure 4 presents example answers generated by different models for a single query. **GPT-4o** chieves the best overall performance across both evaluation scopes, leading on InstAcc, ArgAcc, SummAcc, and on P, O, and Ans. Among open-source large models, **Llama-3-70B-Instruct** ranks highest, with competitive ToolAcc and strong end-to-end results, notably on O and L. For medium-scale open-source models, **Qwen2.5-7B-Instruct** stands out with the highest L score (Spatial Relations) of all models, despite weaker InstAcc and SummAcc. **DeepSeek-V3** and **Qwen2.5-32B-Instruct** deliver moderate results but consistently lag behind commercial models, especially on ArgAcc and AnsAcc. Smaller models such as **Phi-3-Mini**, **Mistral-7B-Instruct**, and **Yi-1.5-6B** show significant drops across metrics, particularly on O and Ans., indicating limited robustness in multi-step tool reasoning. Across all models, **argument consistency** (ArgAcc) is the primary bottleneck, and poor **tool selection** (ToolAcc) strongly correlates with reduced end-to-end accuracy. Even the strongest systems achieve only modest absolute scores, underscoring the difficulty of reliable geospatial reasoning, accurate argument propagation, and faithful summary generation in GeoHOP.

## 5.3 TASK ANALYSIS

Across models, we observe clear and systematic differences in both task success rates and execution dynamics on the GeoHOP benchmark (Figure 5). In Subfigure 5(b), **API-based models** such as **GPT-4o** and **GPT-3.5** achieve the highest numbers of successful task completions,

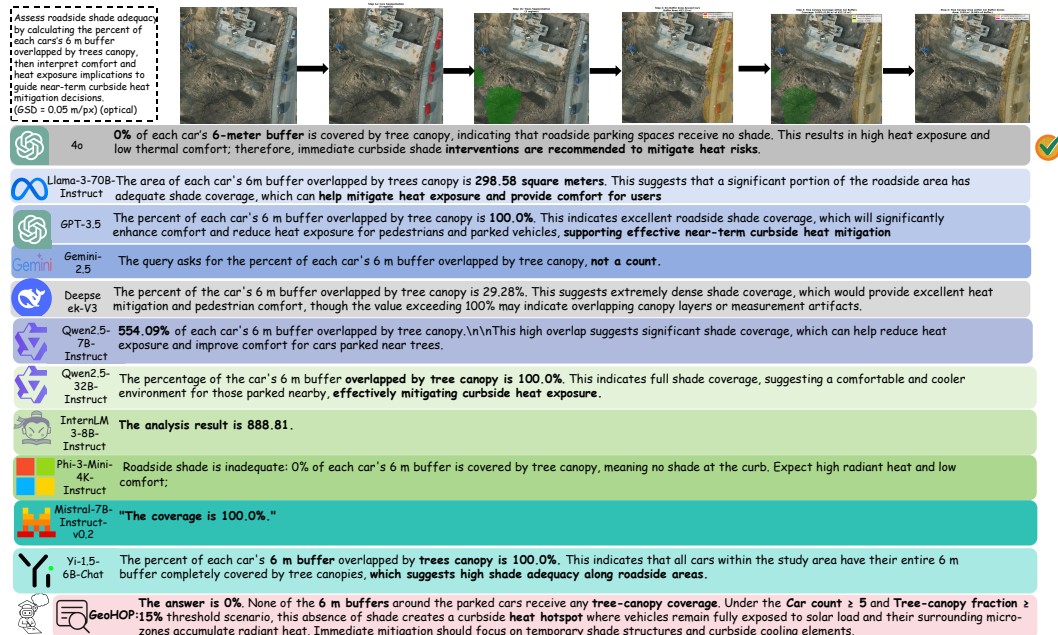

Figure 4: Comparative Evaluation of LLMs on Spatial Reasoning Tasks.

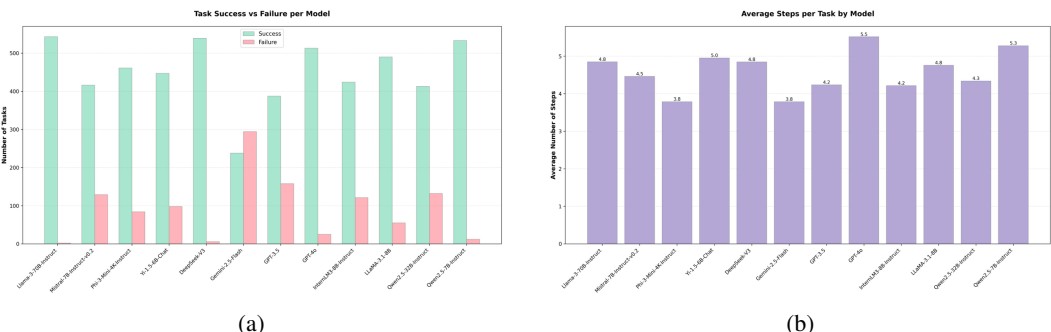

| (a) | (b) |

Figure 5: Comparison of (a) task success vs. failure distribution and (b) average reasoning steps across models on GeoHOP. Frontier API-based models consistently demonstrate higher robustness and deeper tool-augmented reasoning chains, while open-source models exhibit higher failure rates and shallower execution sequences. These patterns highlight GeoHOP's emphasis on multi-step, tool-integrated spatial reasoning and reveal structural differences in agentic planning capabilities across model families.

with markedly fewer failures than open-source systems. By contrast, **Qwen2.5-32B-Instruct**, **Qwen2.5-7B-Instruct**, and **DeepSeek-V3** show much higher failure counts, reflecting weaker instruction adherence and less consistent tool-execution reliability. Subfigure 5(a) examines *reasoning depth*. Large open-source models such as **Llama-3-70B-Instruct** and **Qwen2.5-57B-Instruct** produce longer reasoning chains (4.8–5.3 steps on average), while mid-sized systems like **Yi-1.5-6B-Chat** and **Gemini-2.5-Flash** adopt shallower plans (3.8 steps). **GPT-4o** balances depth and reliability with an average of about 4.2 steps per task, achieving a favorable trade-off between plan length and execution success.

Overall patterns in Figure 5 show that **higher success rates are not simply a product of longer chains**. Strong performance requires *coherent multi-step planning*, robust *argument selection*, and dependable *tool-integrated execution*. **GPT-4o** most consistently balances chain length, execution stability, and task-level success across all evaluated systems.

Table 4: Percentage distribution of four categories of errors for different LLMs.

| Model | Format Error (%) | Reasoning Error (%) | Perception Error (%) | Tool-Use Error (%) |
|---|---|---|---|---|
| GPT-4o | 41.65 | 13.49 | 41.26 | 3.60 |
| Gemini-2.5-Flash | 29.47 | 21.18 | 12.09 | 37.26 |
| GPT-3.5 | 31.20 | 18.60 | 10.70 | 39.50 |
| DeepSeek-V3 | 66.40 | 23.00 | 8.70 | 1.90 |
| Qwen2.5-32B-Instruct | 31.27 | 19.78 | 13.99 | 34.96 |
| Llama-3-70B-Instruct | 55.20 | 25.70 | 18.20 | 0.90 |
| Qwen2.5-7B-Instruct | 42.65 | 23.48 | 29.37 | 4.50 |
| InternLM3-8B-Instruct | 32.70 | 23.90 | 11.40 | 32.00 |
| LLaMA3-1-8B | 38.36 | 24.58 | 20.28 | 16.78 |
| Phi-3-Mini-4K-Instruct | 45.60 | 34.70 | 1.80 | 17.90 |
| Mistral-7B-Instruct-v0.2 | 45.40 | 21.20 | 5.40 | 28.00 |
| Yi-1.5-6B-Chat | 37.90 | 28.60 | 8.60 | 24.90 |

## 5.4 Erro Analysis

When comparing model families, format errors emerge as the most common failure mode, particularly among open-source systems such as **DeepSeek-V3** (66.40%), **Llama-3-70B** (55.20%), and **Phi-3-Mini** (45.60%). Even frontier API models like **GPT-4o** show a 41.65% rate, indicating that producing valid, structured tool calls—especially JSON argument fields—remains the primary bottleneck in geospatial tool-augmented reasoning. Tool-use errors further separate the model classes: **GPT-4o** is most reliable at 3.60%, whereas **Gemini-2.5-Flash** (37.26%) and **GPT-3.5** (39.50%) frequently misuse operators or supply invalid parameters. Low tool-use error scores for some open-source models (**DeepSeek-V3**, **Llama-3-70B**) often reflect conservative behaviour—avoiding invocation of complex tools when uncertain—rather than accurate execution. A second clear trend involves perception-driven failures, which notably impact API models. **GPT-4o** has a 41.26% perception error rate, linked to its reliance on interpreting land-cover masks, change maps, and spatial layouts. Mid-sized open-source models (e.g., **Qwen2.5-7B**, **LLaMA3-8B**) display more moderate perception errors but higher instability in reasoning or tool usage, consistent with avoiding deep perceptual grounding in favour of heuristic shortcuts. Reasoning errors are generally moderate across all systems (13–35%) and do not appear to be the main bottleneck.

Together, these results indicate that robust structured output generation and stable multimodal perceptual grounding—not abstract reasoning—are the key limitations for current geospatial agents, highlighting the need for better format alignment and stronger cross-modal grounding in future systems.

## 6 Conclusion

In this work, we presented **GeoHOP**, the first benchmark explicitly designed for cognitive-level geospatial reasoning across optical, SAR, and IR modalities. GeoHOP comprises 545 hierarchy-aware, scenario-driven tasks that extend beyond perception to demand structured, multi-step reasoning. To address these challenges, we introduced **GeoPlanner**, an LLM-powered agent that organizes analytic tools into functional hierarchies, supports modality-aware planning, and ensures robust error recovery during execution. Extensive experiments on GeoHOP demonstrate that frontier models such as the GPT-4 family currently achieve the strongest performance. Simultaneously, our benchmark reveals significant potential for improvement in argument accuracy, summary generation, and multimodal toolkit integration. These findings indicate that while progress has been made, substantial headroom remains for advancing reasoning, planning, and execution capabilities in real-world remote sensing scenarios. By providing both a high-fidelity benchmark and a fault-tolerant agentic framework, our work establishes a rigorous foundation for evaluating and advancing multimodal reasoning in RS.

## USE OF LLMs

In this work, we employed large language models (LLMs) to assist with language refinement and to enhance the overall coherence of the manuscript. Specifically, LLMs were used to polish sentence-level grammar and improve the logical flow between sections. In addition, we utilized LLMs to generate a set of scalable vector graphics (SVGs), which served as schematic figures to illustrate the conceptual framework of our study. These applications were limited to stylistic editing and visualization support; all research design, data analysis, and substantive conclusions were conducted and validated independently by the authors.

## ETHICS STATEMENT

This work uses only publicly available remote sensing datasets that do not contain personal or sensitive information. All experiments were conducted in compliance with relevant data usage licenses. We acknowledge potential risks of misuse of geospatial AI technologies, such as unauthorized surveillance or environmental misinterpretation, and emphasize that our methods are intended solely for scientific and societal applications, including environmental monitoring and sustainable development. No human subjects were involved in this study.

## REPRODUCIBILITY STATEMENT

To ensure the reproducibility of our results, we provide the complete framework code of the Spatial Reasoning Agent, including tool integration, planning modules, and evaluation scripts.

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

# A APPENDIX

## A.1 FINE-GRAINED TASK ANALYSIS

Table 5: Fine-grained results across models and capability subsets

| Model | Task | Inst.(%) | Tool.(%) | Arg.(%) | Summ.(%) | Ans.(%) |
|---|---|---|---|---|---|---|
| **API-based** | | | | | | |
| GPT-4o | S+G | 77.78 | 92.11 | **32.89** | 75.00 | **45.29** |
| | S+Q | 78.33 | 89.20 | 22.80 | 94.00 | 36.00 |
| | S+G+Q | **85.20** | 84.40 | 27.37 | 91.08 | 35.43 |
| | H ∩ (S+G+Q) | 60.70 | 46.77 | 18.09 | 60.47 | 9.81 |
| Gemini-2.5-Flash | S+G | 49.44 | 48.03 | 18.42 | 53.57 | 21.43 |
| | S+Q | 30.67 | 25.60 | 12.00 | 30.00 | 8.00 |
| | S+G+Q | 34.51 | 23.40 | 9.00 | 24.93 | 9.97 |
| | H ∩ (S+G+Q) | 50.81 | 34.24 | 12.40 | 41.86 | 0.00 |
| GPT-3.5 | S+G | 67.2 | 73.0 | 27.0 | 64.3 | 42.9 |
| | S+Q | 83.3 | 94.4 | 40.8 | 98.0 | **48.0** |
| | S+G+Q | 53.9 | 53.5 | 20.8 | 51.2 | 24.7 |
| | H ∩ (S+G+Q) | 67.1 | 54.4 | **18.3** | 69.8 | 10.5 |
| **Open-source (Large)** | | | | | | |
| DeepSeek-V3 | S+G | 80.0 | 86.8 | 28.3 | **92.9** | 32.1 |
| | S+Q | 83.3 | **96.8** | 30.0 | 98.0 | 40.0 |
| | S+G+Q | 61.9 | 65.7 | 24.0 | 53.8 | 21.3 |
| | H ∩ (S+G+Q) | 70.7 | **54.8** | 17.7 | 60.5 | 5.8 |
| Qwen2.5-32B-Instruct | S+G | 59.4 | 65.1 | 23.7 | 60.7 | 10.7 |
| | S+Q | 83.3 | 94.4 | **41.2** | 98.0 | 32.0 |
| | S+G+Q | 60.1 | 57.5 | 15.4 | 52.0 | 21.3 |
| | H ∩ (S+G+Q) | 66.0 | 46.5 | 15.2 | 66.3 | 5.8 |
| Llama-3-70B-Instruct | S+G | 80.00 | 80.26 | 24.34 | 89.29 | 25.00 |
| | S+Q | **83.33** | 92.80 | 38.80 | 94.00 | 34.00 |
| | S+G+Q | 64.44 | 70.12 | 23.17 | 62.99 | 23.10 |
| | H ∩ (S+G+Q) | 72.21 | 47.67 | 15.37 | **80.23** | 11.63 |
| **Open-source (Medium/Small)** | | | | | | |
| Qwen2.5-7B-Instruct | S+G | 80.0 | **94.7** | 32.2 | 85.7 | 42.9 |
| | S+Q | 83.3 | 96.0 | 32.4 | **100.0** | 34.0 |
| | S+G+Q | 78.7 | 77.7 | 23.5 | 57.5 | 22.0 |
| | H ∩ (S+G+Q) | 69.0 | 53.6 | 17.1 | 73.3 | **12.8** |
| InternLM3-8B-Instruct | S+G | 74.4 | 82.9 | 28.9 | 78.6 | 25.0 |
| | S+Q | 83.3 | 94.4 | **41.2** | **100.0** | 32.0 |
| | S+G+Q | 55.8 | 55.2 | 22.2 | 38.6 | 19.7 |
| | H ∩ (S+G+Q) | 70.2 | 49.6 | 16.8 | 64.0 | 5.8 |
| LLaMA3-1-8B | S+G | 80.0 | 89.5 | 28.3 | 85.7 | 21.4 |
| | S+Q | 51.7 | 51.6 | 28.4 | 40.0 | 6.0 |
| | S+G+Q | 71.2 | 71.0 | 23.2 | 50.7 | 19.2 |
| | H ∩ (S+G+Q) | 67.3 | 48.7 | 16.0 | 76.7 | 10.5 |
| Phi-3-Mini-4K-Instruct | S+G | 51.11 | 42.11 | 14.47 | 32.14 | 7.14 |
| | S+Q | 77.67 | 82.00 | 32.40 | 72.00 | 16.00 |
| | S+G+Q | 57.35 | 40.98 | 18.34 | 2.62 | 2.62 |
| | H ∩ (S+G+Q) | 41.86 | 28.29 | 8.79 | 0.00 | 0.00 |
| Mistral-7B-Instruct-v0.2 | S+G | 73.9 | 74.3 | 24.3 | 78.6 | 25.0 |
| | S+Q | 77.7 | 86.0 | 36.4 | 92.0 | 16.0 |
| | S+G+Q | 56.0 | 43.6 | 15.1 | 37.3 | 13.6 |
| | H ∩ (S+G+Q) | 65.3 | 40.8 | 14.3 | 47.7 | 5.8 |
| Yi-1.5-6B-Chat | S+G | **81.67** | 63.82 | 23.68 | 85.71 | 14.29 |
| | S+Q | 54.33 | 56.40 | 21.20 | 38.00 | 14.00 |
| | S+G+Q | 60.37 | 41.02 | 15.49 | 30.97 | 11.02 |
| | H ∩ (S+G+Q) | **73.37** | 36.82 | 14.60 | 53.49 | 6.98 |

Table 5 organizes the GeoHOP benchmark into four fine-grained task subsets, covering distinct combinations of semantic perception, geometric reasoning, quantitative analysis, and long-horizon planning. This division enables systematic interpretation of model performance along well-defined spatial reasoning dimensions:

- **Semantic + Geometry (S+G)**: Requires semantic understanding plus geometric reasoning (e.g., segmentation followed by buffering or overlap), excluding quantitative tools.
- **Semantic + Quantitative (S+Q)**: Combines semantic perception with numerical measurement (e.g., detection plus distance calculation or counting), without geometric relations.

- **Semantic + Geometry + Quantitative (S+G+Q)**: Full-pipeline workflows integrating all three capabilities (e.g., segmentation → buffering → area measurement), representing the most comprehensive multimodal spatial reasoning.

- **Long-Horizon (H)**: Tasks with four or more sequential tool operations ($K \geq 4$), requiring coherent multi-stage planning, dependency tracking, and error-free orchestration. The subset **H ∩ (S+G+Q)** captures long-horizon tasks that also integrate all three capability types, posing the highest complexity.

**S+G**: **GPT-4o** leads, particularly in **Tool** (92.11%) and **Summ.** (75.00%) accuracy. **DeepSeek-V3** also scores strongly, with **Inst.** at 80.0% and **Summ.** at 92.9%. Conversely, **Qwen2.5-7B-Instruct** drops sharply in more complex settings such as S+G+Q.

**S+Q**: **GPT-3.5** excels, with **Tool** at 94.4% and **Summ.** at 98.0%. **Llama-3-70B** performs similarly in **Tool** (92.8%), but—like GPT-3.5—declines in S+G+Q. **Gemini-2.5-Flash** struggles here, achieving only 25.6% in **Tool**.

**S+G+Q**: **GPT-4o** dominates across categories, with **Inst.** 85.20%, **Tool** 84.40%, and **Summ.** 91.08%, evidencing its strength in synthesizing semantic, geometric, and quantitative reasoning. **DeepSeek-V3** is less consistent, with **Inst.** at 61.9% and **Ans.** only 21.3%.

**H ∩ (S+G+Q)**: This most demanding subset highlights the gap between strong and weaker systems. **GPT-4o** remains competitive in **Tool** (46.77%) and **Summ.** (60.47%), but its **Ans.** is only 9.81%. **Gemini-2.5-Flash** collapses entirely for final answers (**Ans.**=0%), with **Tool** at just 34.24%. Smaller models, such as **Qwen2.5-7B-Instruct**, also falter here, showing steep drops in **Ans.** (12.8%).

**Overall trends**: API-based models, especially **GPT-4o**, maintain strong performance across all subsets, particularly complex integrations like S+G+Q and long-horizon H ∩ (S+G+Q). Open-source models often perform well on simpler S+G and S+Q tasks, but suffer substantial degradation on multi-capability, long-horizon workflows—underscoring the difficulty of sustained, high-fidelity tool reasoning in extended geospatial pipelines.

## A.2 MODALITY-DEPENDENT ACCURACY COMPARISON

| Model | Overall (%) | Optical (%) | SAR (%) | IR (%) |
|---|---|---|---|---|
| GPT-4o | **25.70** | **29.29** | 11.50 | **100.00** |
| GPT-3.5 | 24.40 | 28.05 | 0.00 | 56.82 |
| Gemini-2.5-Flash | 12.04 | 11.63 | 0.00 | **100.00** |
| DeepSeek-V3 | 21.28 | 25.19 | 0.00 | 43.18 |
| Qwen2.5-32B-Instruct | 19.08 | 23.12 | 0.00 | 34.09 |
| Qwen2.5-7B-Instruct | 22.57 | 22.08 | 17.24 | 40.91 |
| Llama-3.1-8B-Instruct | 16.70 | 14.81 | **26.72** | 6.82 |
| Llama-3-70B-Instruct | 22.75 | 27.27 | 0.00 | 43.18 |
| InternLM3-8B-Instruct | 18.90 | 22.08 | 0.86 | 38.64 |
| Mistral-7B-Instruct-v0.2 | 13.21 | 16.36 | 0.00 | 20.45 |
| Phi-3-Mini-4K-Instruct | 3.67 | 0.52 | 8.62 | 18.18 |
| Yi-1.5-6B-Chat | 10.83 | 13.51 | 0.00 | 15.91 |

Table 6: Accuracy comparison across models on the overall benchmark and on each sensing modality (Optical, SAR, IR).

As shown in Table 6, GeoHOP performance exhibits strong modality-dependent variation. Frontier API models lead overall, with **GPT-4o** achieving the highest benchmark accuracy and excelling in both Optical (29.29%) and IR (100%) tasks. Yet even GPT-4o drops sharply on SAR (11.50%), underscoring that radar-domain reasoning remains challenging for current LLMs.

Open-source models show highly uneven behavior: some (e.g., **DeepSeek-V3**, **Qwen2.5-32B**) are competitive on Optical but collapse on SAR, with seven models scoring 0%. **Llama-3.1-8B** is the most robust in this domain, posting the highest SAR accuracy (26.72%), suggesting comparatively stronger resilience to speckle-dominated radar imagery.

| Model | Optical Accuracy (%) |
|---|---|
| **RS-MLLMs** | |
| EarthDial(Soni et al., 2025) | 9.38 |
| GeoChat(Kuckreja et al., 2023) | 6.70 |
| GeoPix(Ou et al., 2025) | 6.17 |
| **LLM-based Agent (Ours)** | |
| **GeoPlanner (GPT-4o)** | **29.29** |

Table 7: Optical accuracy comparison between representative remote-sensing multimodal LLMs (RS-MLLMs) and our **GeoPlanner (GPT-4o)** on GeoHOP optical VQA tasks.

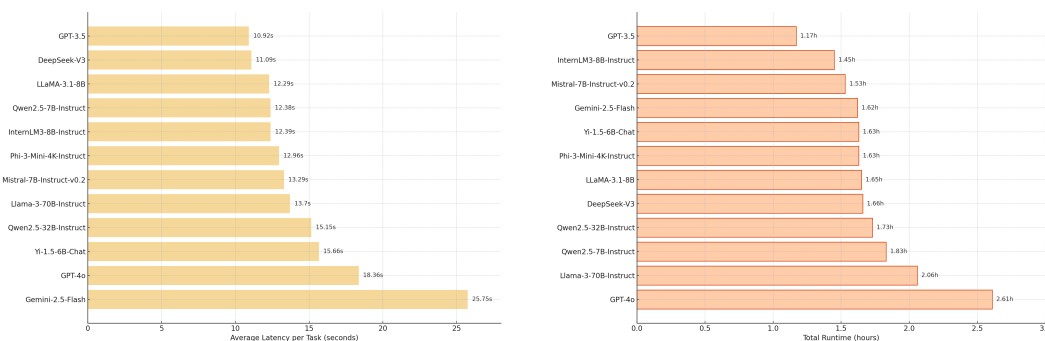

(a) Per-task inference latency across evaluated models. Models with faster single-task inference exhibit lower reasoning overhead.

(b) Total runtime required to complete the full 545-task GeoHOP benchmark. Results highlight cumulative computational efficiency at scale.

Figure 6: Comparison of model-level computational efficiency. Left: average latency per task. Right: full-benchmark total runtime.

IR performance is sharply polarized—frontier API models achieve perfect accuracy, while smaller open-source systems remain below 45%—indicating that IR image interpretation benefits more from high model capacity than from cross-modal generalization.

Overall, these results confirm that accuracy disparities across Optical, SAR, and IR are systematic, and that geospatial reasoning performance is constrained as much by sensing modality as by model scale and architecture.

### A.3    COMPARISON WITH REMOTE SENSING MULTIMODAL LLMs

As shown in Table 7, existing **remote-sensing multimodal LLMs (RS-MLLMs)** perform poorly on the GeoHOP benchmark's optical geospatial reasoning tasks. Models such as **EarthDial** (9.38%), **GeoChat** (6.70%), and **GeoPix** (6.17%) all struggle to exceed even 10% accuracy, despite being specifically designed for remote-sensing understanding.

In contrast, our **GeoPlanner (GPT-4o)** achieves a substantially higher accuracy of **29.29%**, highlighting its stronger ability to integrate semantic cues, spatial relations, and tool-based execution. Nevertheless, the overall performance levels—both for RS-MLLMs and GeoPlanner—indicate that GeoHOP's optical tasks remain highly challenging, requiring models to handle fine-grained spatial reasoning beyond standard VQA-style interpretation.

### A.4    COMPUTATIONAL COST ANALYSIS

Figures 6 and 7 present a detailed comparison of computational costs across models on the GeoHOP benchmark, measured by per-task inference latency, total runtime, and tool-usage distribution.

**Per-task latency** reflects single-task completion speed. **GPT-3.5** and **DeepSeek-V3** are the fastest, averaging 10.92 s and 11.09 s per task, respectively. In contrast, **Gemini-2.5-Flash** (25.75 s) and **GPT-4o** (18.36 s) have considerably higher latencies, indicating more extensive reasoning overhead.

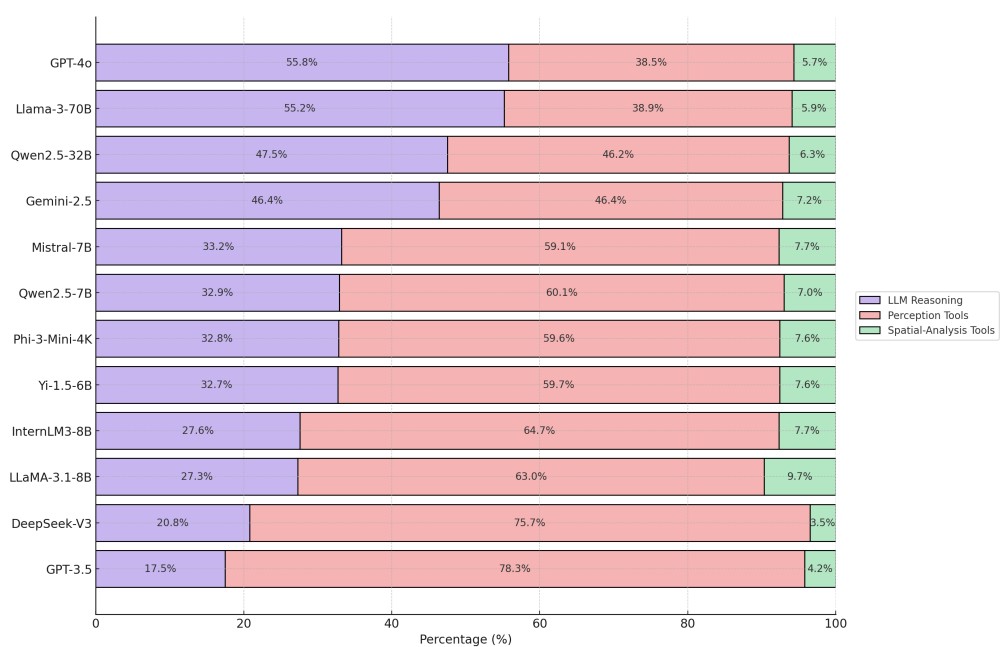

Figure 7: Breakdown of model tool usage across the 545-task GeoHOP benchmark. Each bar shows the relative proportion of **LLM reasoning**, **perception tool calls**, and **spatial-analysis operators**. Higher-performing models tend to rely more on LLM-driven reasoning, whereas less capable models depend more heavily on perception and spatial-analysis tools.

**Total runtime** over all 545 tasks highlights cumulative efficiency: **GPT-3.5** completes the benchmark in 1.17 h, compared with 2.61 h for **GPT-4o**, which reflects the trade-off between speed and richer multi-step reasoning.

**Tool-usage breakdown** (Figure 7) shows the proportion of execution devoted to *LLM reasoning*, *perception tools*, and *spatial-analysis tools*. Higher-performing systems, such as **GPT-4o** and **Llama-3-70B**, allocate over 55% of their execution to LLM-driven reasoning, while models like **DeepSeek-V3** and **GPT-3.5** show heavier reliance on perception and spatial-analysis operators.

These results emphasize the importance of optimizing tool-invocation strategies and striking a balance between reasoning depth and computational efficiency, which is essential for scalable deployment of spatial reasoning agents on large-scale geospatial workloads.

A.5    BENCHMARK COMPARISON OVERVIEW

To situate GeoHOP within the broader remote-sensing benchmark landscape, Figure 8 shows representative examples from systems discussed in our Related Work section. Benchmarks such as *Tree-GPT*, *Change-Agent*, *Remote Sensing ChatGPT*, *RS-Agent*, *GeoLLM-Engine*, *UnivEARTH*, and *ThinkGeo* primarily target perception-centric capabilities, including tree-attribute extraction, bitemporal change detection, object detection, and natural-language querying over curated datasets.

Although these systems showcase notable progress in specialized RS tasks, their workflows are typically confined to single-step or shallow reasoning pipelines. In contrast, the GeoHOP benchmark (bottom example in Figure 8) defines cognitively grounded, multi-step geospatial reasoning challenges that demand integration of perception, spatial-relation operators, and spatial-statistics tools. This comparison underscores the conceptual gap between conventional RS benchmarks and the tool-augmented, computation-verifiable reasoning workflows evaluated in GeoHOP.

A.6    SCENE-CONTEXT KNOWLEDGE BASE

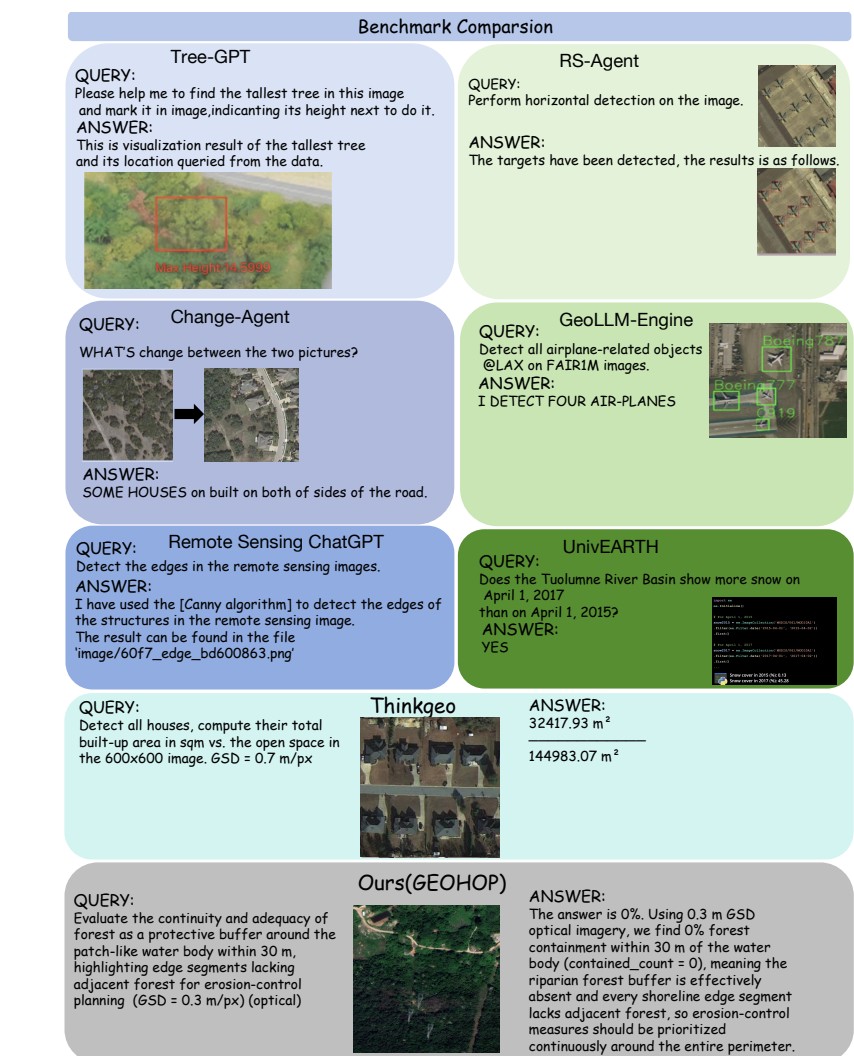

Figure 8: **Examples of representative remote-sensing benchmarks discussed in our Related Work.** This figure summarizes characteristic task formats from several established RS benchmarks—including *Tree-GPT*, *Change-Agent*, *Remote Sensing ChatGPT*, *RS-Agent*, *GeoLLM-Engine*, *UnivEARTH*, and *ThinkGeo*—covering tree-parameter extraction, change detection, object detection, semantic interpretation, and natural-language-based geospatial querying. For comparison, our *GeoHOP* benchmark (bottom row) introduces cognitively grounded, tool-augmented geospatial reasoning tasks requiring multi-step computation, spatial relations, and actionable environmental interpretation.

To ground generation in domain knowledge, we inject a compact knowledge corpus into prompts. The corpus covers four guidance categories: (i) urban greening and heat-mitigation frameworks (Twohig-Bennett & Jones, 2018; Bowler et al., 2010; Aram et al., 2019; Norton et al., 2015; Rigolon, 2016); (ii) international land-cover standards (Di Gregorio & Jansen, 1998; Mosca et al., 2020); (iii) aviation and maritime search-and-rescue doctrine for IR small-target tasks (Kim et al., 2020); and (iv) industrial safety rules for separation and proximity of hazardous assets (Ricci et al., 2021; Kukfisz et al., 2022).

| Category | Scene Context Type | Geometric / Relational Elements | Example Question |
|---|---|---|---|
| StormWater Runoff Hotspot | Urban zones with high building coverage and minimal vegetation, indicating strong surface-runoff potential, limited infiltration capacity, and elevated pluvial-flood and wash-off risk. | AOI-level class area fractions: Building ≥ 25% and Low vegetation ≤ 20%. | Assess stormwater runoff hotspot potential by segmenting building and low vegetation, then buffering building edges by 5 m to measure the proportion of low vegetation within this zone and interpret whether buildings are clustered with fragmented adjacent low vegetation. (GSD = 0.09 m/px) (optical) |
| Agro--Urban First-Flush Hazard Zone | Peri-urban agro-urban mosaics where substantial cropland and emerging built-up areas share the same watershed, creating strong first-flush nutrient and pesticide pulses and elevated pluvial-flood and combined-sewer-overflow risk during storms. | AOI-level fractions: Agriculture ≥ 30%, Building ≥ 10%. Geometric Relation: Buildings within a 50 m buffer of Agriculture field edges. | How many Buildings lie within 50 m of Agriculture fields, quantifying settlement encroachment along field edges in an agro-urban first-flush zone? (GSD = 0.3 m/px) (optical) |
| Forest--Settlement Cooling Fringe Micro-Zone | Low-density or emerging settlements located along forest margins where nearby tree-covered areas provide local cooling, shading, and greening benefits to residents under hot weather, consistent with urban greening and heat-mitigation frameworks. | Distance-based cooling-access band between Buildings and Forest edges: near-forest cooling influence when D(Building, Forest) ≤ 60 m. | What proportion of building area lies within 60 m of forest edges, indicating micro-scale access to cooling benefits from adjacent forests? (GSD = 0.3 m/px) (optical) |
| Built--Tree Street Canyon Heat--Pollution Trade-off Zone | Urban street fabrics with partial tree canopy where shading can mitigate heat, but in confined canyons may also hinder pollutant dispersion and exacerbate near-surface air-quality issues under low-wind, high-temperature conditions. | AOI-level class area fractions: Building ≥ 15% and Trees ≥ 10%. | Assess urban heat island intensity by analyzing whether the dominant building block is highly concentrated and whether surrounding trees cover is sparse or fragmented; compute area proportions of building versus trees to support the interpretation. (GSD = 0.09 m/px) (optical) |
| Waterfront Blue-Space Cooling and Access Fringe | Urban residential or mixed-use developments located directly along major rivers, lakes, bays, or tidal channels, where close proximity to surface water provides local cooling, breeze exposure, and blue-space health benefits under hot weather, in line with urban greening and blue-space frameworks. | Distance-based blue-space exposure threshold between Buildings and Water bodies: near-waterfront exposure when D(Building, Water) ≤ 50 m. | How many buildings lie within 50 m of water bodies, indicating micro-scale access to cooling and blue-space exposure along the waterfront? (GSD = 0.3 m/px) (optical) |
| Forest--Bare Ground Degradation Edge | Forested areas mapped as Land Cover Classification System tree-covered classes but bordered by nearby bare or recently cleared land, where exposed forest boundaries are recognised in UNCCD land-degradation guidance as loci of increased edge stress, connectivity loss, and erosion risk. | Distance-based degradation-screening threshold between Barren land and Forest edges: edge-degradation relevant when D(Barren, Forest) ≤ 30 m. | Do barren patches intersect the 30 m forest-edge belt, indicating forest boundaries exposed to bare-ground degradation pressure? (GSD = 0.3 m/px) (optical) |
| Settlement--Bare Ground Dust-Exposure Fringe | Populated dryland or peri-urban areas where residential buildings border extensive bare or recently cleared land mapped as Land Cover Classification System bare areas, identified in United Nations Convention to Combat Desertification land-degradation and sand-and-dust-storm guidance as zones of elevated wind-blown dust exposure for nearby communities. | Distance-based dust-exposure screening threshold between Buildings and Barren areas: dust-exposure relevant when D(Building, Barren) ≤ 40 m. | How many buildings lie within 40 m of barren land, indicating potential near-source exposure to wind-blown dust from adjacent bare surfaces? (GSD = 0.3 m/px) (optical) |

Figure 9: **Scene-context types for urban greening and heat-mitigation frameworks.** This figure summarizes domain knowledge related to stormwater runoff, cooling-access fringes, dust-exposure belts, blue-space interfaces, and street-canyon heat–pollution trade-offs, formalized via simple spatial thresholds and adjacency relations.

| Category | Scene Context Type | Geometric / Relational Elements | Example Question |
|---|---|---|---|
| Urban--Rangeland Wildland--Urban Interface--Grass Interface | Peri-urban landscapes where built-up areas abut extensive rangeland, creating a grass-fuel wildland--urban interface with elevated ignition and spread potential. | Geometric Relation: Urban land within 30 m of Rangeland (high-risk Wildland--Urban Interface adjacency) and within 30--100 m of Rangeland (intermediate Wildland--Urban Interface belt). | For the Urban--Rangeland Wildland--Urban Interface--Grass Interface, what share of Rangeland lies near Urban land within the interface region, measuring the degree of adjacency relevant to Wildland--Urban Interface management? (GSD = 0.50 m/px) (optical) |
| Rangeland--Bare Ground Degradation Mosaic | Dryland rangeland mapping units where grazing land coexists with substantial bare ground, forming a mixed Land Cover Classification System class that signals emerging land degradation and heightened wind and water erosion risk along a desertification front. | AOI-level fractions: Rangeland >= 40%, Barren land >= 20%. Geometric Relation: 50 m buffer around Barren land intersected with Rangeland to quantify exposure. | What percentage of Rangeland area falls within a 50 m buffer around Barren land, indicating the degree of rangeland exposure to the advancing desertification front? (GSD = 0.50 m/px) (optical) |
| Forest--Rangeland Ecotone Fire--Propagation Belt | Mixed forest--grass mosaics representing forest--rangeland ecotones where substantial cover from both classes creates a potential corridor for grassfire spread into wooded patches. | AOI-level fractions: Forest >= 30%, Rangeland >= 30%; Geometric Relation: Forest land within a 60 m buffer of Rangeland edges. | What percentage of Forest land lies within a 60 m buffer of Rangeland edges, estimating forest exposure to grassfire spread along the ecotone? (GSD = 0.50 m/px) (optical) |
| Agro--Rangeland Mixed Production Mosaic | Integrated crop--pasture mosaics where cropland and grazing land are managed jointly, and field--pasture boundaries act as key zones for nutrient transfer, erosion, and runoff control in mixed production systems. | AOI-level fractions: Agriculture land >= 40%, Rangeland >= 20%; Geometric Relation: Agriculture land within a 50 m buffer around Rangeland polygons. | What percentage of Agriculture land lies within a 50 m buffer around Rangeland polygons, quantifying the crop--pasture interface and its exposure to edge-of-field runoff in a mixed production mosaic? (GSD = 0.50 m/px) (optical) |
| Agro--Soil Degradation Hotspot | Cropland landscapes where extensive bare soil patches within predominantly agricultural areas indicate elevated erosion risk, accelerated topsoil loss, and declining soil productivity under non-conservation management. | AOI-level fractions: Agriculture >= 50%, Barren >= 30%; Geometric Relation: Barren land fully enclosed by Agriculture polygons, delineating in-field degradation patches. | What percentage of Barren land area is fully enclosed by Agriculture land, identifying in-field soil degradation patches that are priority targets for site-specific restoration in cropland fields? (GSD = 0.3 m/px) (optical) |
| Agro--Transport Sediment Connectivity Corridor | Cropland landscapes intersected by road networks that act as efficient conduits, enhancing runoff and sediment-transport connectivity between erodible fields and drainage channels. | AOI-level fractions: Agriculture >= 50%, Road >= 5%; Geometric Relation: Agriculture land within 30 m of Roads, representing road-linked sediment pathways from fields to the drainage network. | What percentage of Agriculture land lies within 30 m of Roads, indicating how strongly fields are coupled to transport corridors for access and sediment delivery? (GSD = 0.3 m/px) (optical) |
| Agro--Dominated Lentic HAB and Hypoxia Risk Zone | Agricultural landscapes containing small lentic water bodies that can retain runoff-borne nutrients, creating conditions for harmful algal blooms (HABs) and hypoxic events that threaten drinking-water security and aquatic ecosystems. | AOI-level fractions: Agriculture-dominated landscapes with mapped Water bodies; Relation: Water-body shorelines within 30 m of Agriculture land, representing runoff-critical zones inside the nominal buffer width. | What percentage of Water-body area lies within 30 m of Agriculture land, indicating lentic ponds most exposed to agrochemical runoff and HAB/hypoxia risk? (GSD = 0.3 m/px) (optical) |
| Dryland Road Dust Corridor | Degraded dryland surfaces with extensive bare or sparsely vegetated land adjoining primary transport routes, where United Nations Convention to Combat Desertification type bare areas act as dust-source zones and vehicle traffic and wind enhance fugitive dust emissions, soil loss, and local air-quality impacts. | AOI-level class area fractions: Barren >= 30% and Road >= 5%. | What percentage of barren land lies within 30 m of roads, as a proxy for dust-source corridors along transport routes? (GSD = 0.3 m/px) (optical) |
| Shrinking Lakebed Dust and Turbidity Hotspot | Shoreline zones of shrinking lakes or reservoirs where exposed barren lakebed or shoreline soils lie adjacent to remaining open water, identified in Land Cover Classification System land-degradation and sand-and-dust-storm guidance as potential dust-emission and turbidity hotspots. | AOI-level class area fractions: Barren >= 30% and Water >= 10%. | Does barren shoreline overlap more than 10% of water bodies, to flag shrinking-lake dust and turbidity hotspots? (GSD = 0.3 m/px) (optical) |
| Partial--Canopy Parking Heat and Emission Recirculation Hotspot | Parking or traffic areas with partial tree canopy, where large paved surfaces, heat storage, and tailpipe emissions combine with incomplete shading to create elevated thermal and pollution exposure under hot, low-wind conditions. | Object-count and canopy thresholds: Car count >= 5 and Tree canopy fraction >= 15%. | How many cars lie within a 7 m buffer of trees canopy along the central roadway to estimate roadside shade availability? (GSD = 0.05 m/px) (optical) |

Figure 10: **Scene-context types for international land-cover standards (Part I).** This figure presents the first group of LCCS-aligned scene-context types, each defined using area fractions, buffer relations, and proximity thresholds grounded in land-degradation and land-cover classification guidance.

| Forest--Road Ignition Edge Corridor | Forested landscapes mapped as Land Cover Classification System tree-covered areas but intersected by access roads, where even low road fractions create extensive forest edge bands that concentrate human-caused fire ignitions along transport corridors. | AOI-level class area fractions: Forest >= 40% and Road >= 1%. Geometric Relation: Forest within a 25 m buffer of roads as the near-road ignition edge band. | What percentage of forest area lies within 25 m of roads, indicating ignition-prone forest edge along access routes? (GSD = 0.3 m/px) (optical) |
|---|---|---|---|
| Industrial Access Causeway--Flood Isolation Corridor | Low-lying causeway and bridge corridors that provide primary road access to hazardous industrial sites across major water bodies, where small water-road separations increase the risk that flood or surge overtopping will isolate evacuation and emergency-response routes, in line with Natech flood-domino safety guidance. | Distance-based flood-exposure bands between Water patches and Roads: high-risk exposure when D(Water, Road) <= 30 m. | What is the minimum distance from the causeway road to the nearest water patch, to flag flood-exposed access corridors serving hazardous industrial sites? (GSD = 0.3 m/px) (optical) |
| Building Exposure Zone along Hazardous Materials Transportation Corridors | Buildings located within 30 m of major roads, forming a proximity-based exposure zone with elevated traffic-related and hazardous-materials risk. | Distance- and safety-based control band between Buildings and major Roads: risk-critical when D(Building, Road) <= 30 m. | How many buildings are positioned near the transport corridor interface, characterizing their accessibility and exposure around the key junction area? (GSD = 0.3 m/px) (optical) |
| Agro--Forest Shelterbelt and Edge-Protection Zone | Agricultural fields adjoining Land Cover Classification System-mapped forest or tree-covered areas, where near-edge agro--forest strips function as shelterbelts that can reduce wind erosion, protect crops, and moderate microclimatic stress along field margins, in line with FAO agroforestry and conservation agriculture guidance. | Distance-based agro--forest edge-protection threshold between Agriculture and Forest: edge-protection relevant when D(Agriculture, Forest) <= 30 m. | What is the total area of agriculture within 30 m of forest edges, indicating the extent of potential shelterbelt and edge-protection interface along cropland margins? (GSD = 0.3 m/px) (optical) |
| Forest--Water Riparian Buffer Integrity Zone | Watersheds where Land Cover Classification System-mapped forest or tree-covered areas form near-shore riparian buffers along lakes and rivers, providing bank stabilisation, shading, and habitat connectivity at the land-water interface. | Distance-based riparian-buffer threshold between Forest and Water bodies: riparian-buffer relevant when D(Forest, Water) <= 30 m. | How much forest area lies within 30 m of water bodies, indicating the extent of riparian buffer along lake and river margins? (GSD = 0.3 m/px) (optical) |
| Urban--Forest Peri-urban Cooling-Buffer Erosion Zone | Peri-urban landscapes where Land Cover Classification System-mapped forests or tree-covered areas form narrow cooling buffers around built-up zones, and where incremental edge-driven urban expansion converts these forest belts into artificial surfaces, eroding microclimate regulation and heat-mitigation benefits documented for urban greening. | Geometric Relation: Forest areas that converted to artificial surfaces between T1 and T2 and lie within a 100 m buffer of pre-existing artificial surfaces at T1, capturing edge-driven urban expansion into former peri-urban cooling buffers. | Among the areas that changed from forest (T1) to artificial surfaces (T2) between T1 and T2, how much of this converted area lies within 100 m of pre-existing artificial surfaces at T1, to characterize edge-driven urban expansion and loss of cooling-forest buffers? (GSD = 0.45 m/px) (optical) |
| Agro--Wetland Riparian Buffer Expansion Corridor | Agricultural watersheds where cropland is being converted into small ponds or riparian wetland strips near existing water bodies, expanding blue--green buffers that retain nutrients and attenuate floods. | AOI fractions: Agriculture_T1 >= 40%, Water_T2 >= 10%; Geometric Relation: Agriculture-to-Water conversion pixels whose T1 Agriculture locations lie within 30 m of pre-existing Water bodies. | What percentage of Agriculture-to-Water conversion area lies within 30 m of pre-existing Water bodies at T1, indicating cropland being repurposed into near-channel ponds or riparian wetland strips that expand the blue--green buffer? (GSD = 0.45 m/px) (optical) |

Figure 10: **Scene-context types for international land-cover standards (Part II).** Continued from Part I, this page includes additional LCCS-based contexts that formalize forest–road ignition edges, riparian buffers, industrial isolation risks, and agro–forest protection zones using distance- and topology-based reasoning.

| Category | Scene Context Type | Geometric / Relational Elements | Example Question |
|---|---|---|---|
| Harbor--Bridge Coastal Flood Vulnerability Zone | Coastal corridors where harbor basins and bridge structures co-occur in low-lying zones, forming critical transport links highly exposed to sea-level rise, storm-surge flooding, and wave-driven damage. | Object-level class counts: $N\_Harbor \geq 1$ AND $N\_Bridge \geq 1$. | How many bridge footprints encroach on the riparian spatial proximity adjacent to harbor shorelines, indicating structures most exposed to coastal flood and storm-surge impacts? (3 m/px) (sar) |
| Petrochemical–Transport Escalation Zone | High-density industrial corridor where transportation bridges and large-scale petrochemical tanks occur in close spatial proximity, indicating cascading-failure potential. | Object-level class counts: $N\_Bridge \geq 1$ AND $2 \leq N\_Tank \leq 3$. | How many detected storage tanks overlap the public-way spatial proximity surrounding bridge approaches? (3 m/px) (sar) |

Figure 11: **Industrial safety & hazardous-asset proximity.** This figure summarizes scene-context types derived from industrial safety domain knowledge, focusing on how hazardous facilities, transport corridors, coastal structures, and surrounding built environments interact under proximity, exposure, and separation constraints. These contexts formalize risk-relevant spatial configurations—such as asset-to-corridor distances, isolation zones.

| Category | Scene Context Type | Geometric / Relational Elements | Example Question |
|---|---|---|---|
| Infrared Anomalous Emission Zone | Regions where a single infrared small target is detected, indicating a localized thermal anomaly or early-stage event. | AOI-level class area fractions: Target = 1. | Are at least two infrared small targets detected within a 25-pixel-high central corridor (y=100-125, full width) to assess low-altitude movement along the valley floor? (ir) |
| Infrared Clustered Activity Field | Areas where multiple infrared small targets are detected, indicating coordinated or large-scale thermal activity. | AOI-level class area fractions: Target >= 2. | Is there exactly one infrared small target detected in each of the left, center, and right vertical thirds of the IR scene, verifying sector coverage during perimeter surveillance? (ir) |

Figure 12: **Aviation & maritime small-target operational doctrine.** This figure summarizes scene-context types derived from aviation and maritime domain knowledge concerning the behavior, distribution, and interpretability of small or dispersed targets in open-water and open-air environments. These contexts capture recurrent operational patterns—ranging from isolated point-like cues to concentrated activity fields—and express them as measurable spatial configurations. By formalizing these domain-driven patterns into explicit spatial rules and count-based constraints, the scene-context types provide structured guidance for identifying, prioritizing, and interpreting small-target activity.

## A.7 LIMITATIONS AND FUTURE WORK

In this research, our tool-augmented agent provides fine-grained metrics on GeoHOP, advancing the systematic evaluation of geospatial reasoning. Recent advances in code-based agents have shown strong potential for general-domain reasoning. Compared with traditional tool-augmented agents, these code-driven systems offer greater flexibility, richer iterative planning, and the ability to compose new computational procedures—making them an increasingly influential direction in agent research.

To understand how these emerging agents behave in the RS setting, we conducted a direct comparison between a **code-based agent** (Cursor/MCP + RemoteSAM(Yao et al., 2025)) and our **tool-augmented agent**(GeoPlanner) on the detection subset of GeoHOP. As shown in Table 8, the code-based agent achieves higher final answer accuracy (44.63% vs. 40.11%), confirming its advantage in high-level reasoning. However, the tool-augmented agent retains clear strengths that are critical for real-world remote sensing. It demonstrates far superior execution stability ($O = 99.23\%$ vs. 56.50%), reflecting the reliability of pre-verified domain tools. Moreover, tool-augmented agents also provide strong controllability and avoid unsafe or hallucinated operations—factors essential in safety-sensitive RS pipelines, where algorithms such as atmospheric correction or SAR filtering must behave predictably.

These complementary properties suggest that neither paradigm is sufficient on its own. Thus, a promising direction for future RS agents is a *Hybrid Design* that integrates: (i) code-based agent-style code generation for flexible planning and adaptive reasoning, and (ii) tool-augmented agent-style verified tools for critical, high-precision RS operations. We outline this direction as our primary future work, aiming to combine the reasoning power of code-based agents with the operational robustness required for RS deployment.

Table 8: Comparison of Code-based Agent (CBA, i.e., Cursor) and Tool-Augmented Agent (TAA) on the Detection Task Subset. ($\mathcal{O}$ represents the accuracy of spatial statistics tool calls; $\mathcal{L}$ represents the accuracy of spatial relations tool calls; and Ans. represents the final answer accuracy.)

| Agent Type | $\mathcal{O}$ (Operation (%)) | $\mathcal{L}$ (Logic (%)) | Ans. (Accuracy (%)) |
|---|---|---|---|
| **Ours (Tool-Augmented Agent)** | **99.23** | **86.51** | 40.11 |
| **Cursor (Code-based Agent)** | 56.50 | 85.88 | **44.63** |

Table 9: Ablation study comparing the full GeoPlanner framework against ThinkGeo (a Standard ReAct baseline) using GPT-4o. **Inst.**, **Tool.**, **Arg.**, and **Summ.** denote step-by-step accuracies, while **P**, **O**, **L**, and **Ans.** represent end-to-end execution and reasoning scores.

| Method | Step-by-Step Metrics (%) | | | | End-to-End Metrics (%) | | | |
|---|---|---|---|---|---|---|---|---|
| | Inst. | Tool. | Arg. | Summ. | P | O | L | Ans. |
| Thinkgeo(Shabbir et al., 2025)(ReAct) | 77.46 | 48.11 | 14.18 | 46.97 | 48.05 | 5.98 | 50.87 | 8.62 |
| **GeoPlanner** | **79.40** | **72.01** | **23.57** | **82.83** | **72.73** | **88.38** | **77.95** | **25.70** |

## A.8 ABLATION STUDIES

To assess the contribution of GeoPlanner's core architectural components—namely the Hierarchical Tool Organization and Error-Aware Replanning—we conducted an ablation study comparing our full framework against a standard ReAct baseline using the same backbone model (GPT-4o). The ReAct baseline operates with a flat tool list (all operators provided in a single context window, without hierarchical indexing) and relies solely on a linear Thought → Action → Observation loop, lacking GeoPlanner's type-checking, failure-recovery, and replanning modules.

Table 9 summarizes the results on the GeoHOP benchmark. The hierarchical tool organization in GeoPlanner yields notable gains by enabling more accurate and efficient tool retrieval than the flat ReAct toolset. Even more substantial improvements appear in the end-to-end execution metrics, demonstrating the importance of our error-aware replanning module. GeoPlanner proactively detects and recovers from tool-selection and argument errors, maintaining workflow continuity and significantly boosting final answer accuracy.

These ablation results confirm that GeoPlanner's structured architecture is essential for robust geospatial reasoning and cannot be replicated by standard ReAct-style agents.

