# OpenReview forum: "Hierarchies over Pixels: A Benchmark for Cognitive Geospatial Reasoning for Agents"
_ICLR.cc/2026/Conference — Submitted to ICLR 2026_

### Official Review · Reviewer_CA8L · 2025-10-23

**Soundness:** 3
**Presentation:** 3
**Contribution:** 2
**Rating:** 4
**Confidence:** 4

**Summary:**

In this study, the authors address the lack of evaluation benchmarks for assessing the spatial reasoning capabilities of agents in the remote sensing domain by proposing GeoHOP—a benchmark tailored for hierarchical geospatial reasoning. GeoHOP comprises 417 scene-driven, hierarchy-aware tasks covering optical, synthetic aperture radar (SAR), and infrared (IR) imagery, with applications including disaster vulnerability assessment, urban heat island analysis, and forest fragmentation dynamics. Additionally, the authors introduce a simple agent, GeoPlanner, demonstrating its ability to help large language models (LLMs) perform more robust analysis.

**Strengths:**

This paper tackles a valuable gap in the remote sensing community—the lack of novel datasets for evaluating agents—by curating a substantial collection of remote sensing data and using a semi-supervised approach to generate a dataset of 417 tasks spanning multiple modalities (optical, SAR, and IR) for agent evaluation in remote sensing.

**Weaknesses:**

I believe the problem the authors aim to address is highly valuable, and I encourage them to continue this line of work. However, the current results are significantly incomplete, primarily due to the following issues:

1. The authors do not provide a clear definition of what constitutes a "reasoning task" in geospatial contexts. Based on the provided examples, most tasks appear to be multi-turn tool-calling "information retrieval" tasks. Is "complex information retrieval" equivalent to reasoning? This is questionable. In fact, GeoHOP lacks high-difficulty tasks comparable to those in benchmarks like SWE-Bench or BrowseComp. It also misses truly open-ended tasks such as "assess landslide risk in a given region" or "predict local crop yields by integrating diverse data sources," which would require genuine complex reasoning.

2. GeoHOP lacks a discussion of its evaluation framework. Evaluating agent capabilities should probe the boundaries of those capabilities—e.g., through clearly defined evaluation objectives such as task planning, contextual awareness, multimodal understanding, self-correction, etc. Each task should explicitly target specific capabilities, rather than being grouped merely by application scenario as currently done.

3. The paper lacks rigorous assessment of data quality. For a dataset-focused paper, a critical experiment involves evaluating data quality and providing scores and uncertainty estimates for each data instance. Geospatial reasoning may involve both open-ended and closed-ended questions, and different evaluation metrics are needed for each type.

4. GeoHOP is already outdated relative to state-of-the-art agent frameworks in 2025 and does not leverage recent advances in LLM evaluation (e.g., ReAct is an obsolete paradigm). As shown in Figure 1, the so-called "reasoning tasks" are merely simple multi-step tool-calling tasks. While such tasks might have posed a challenge for agents in early 2024, by 2025, general-purpose agents outside remote sensing—such as Claude Code, Gemini CLI, and Cursor—have made significant progress. When integrated with domain-specific protocols like MCP for remote sensing, these agents can already execute complex, multi-faceted tasks. Current open-source or commercial agents can invoke dozens of tools, handle context lengths of 200–300k tokens, and possess self-correction capabilities based on execution feedback. Consequently, the tasks in GeoHOP may be too simplistic for today’s most advanced agents.

**Questions:**

1. What constitutes "reasoning" in geospatial contexts, and what distinguishes "deep reasoning"? How can we measure the difficulty of a reasoning task?

2. How should we separately evaluate the difficulty and answer accuracy for closed-ended versus open-ended geospatial reasoning problems?

3. How does GeoHOP quantify an agent’s performance across various dimensions (e.g., domain knowledge, hallucination, contextual understanding, self-correction)?

4. Can GeoHOP’s data generation pipeline be built entirely within a simulated environment to support reinforcement learning fine-tuning of LLM agents?

---

> ### Author Response · Authors · 2025-11-26
> **Response to Weakness/Questions 1:**
>
> We thank the reviewer for the insightful comments highlighting the need for clearer definitions of geospatial reasoning, a more explicit evaluation framework, a clarification of task difficulty, and guidance on simulation-based data generation. We have revised the manuscript accordingly, and our detailed responses are provided below.
> ### **Clarifying the Definition of Geospatial Reasoning**
>
> In **Section 3.3(Page 5)**, we formalize geospatial reasoning tasks as consisting of:
>
> - **A spatial trigger** – specific geographic context in imagery.
> - **An expert semantic interpretation** – contextual analysis of scene elements.
> - **A verifiable multi-step tool-chain** linking perception to spatial logic.
>
> **Figure 1(Page 4)** has been revised to visualize this hierarchical process, showing query interpretation, semantic retrieval, multi-step planning, parameter inference, and execution feedback.
>
> We added **Appendix A.6 (Page 20)**, which provides domain-knowledge–grounded scene-context examples, including open-ended tasks (e.g., *Built–Tree Street Canyon Heat–Pollution Trade-off analysis*) that require contextual interpretation beyond simple tool calls. This operational definition covers both closed-form and open-form geospatial reasoning.
>
> ---
> ### **Establishing a Comprehensive Evaluation Framework**
>
> The revised manuscript clarifies that our evaluation framework targets specific cognitive capabilities:
>
> - **Task Planning** – sequencing and composing multi-tool workflows
> - **Contextual Awareness** – grounding reasoning in scene context and spatial triggers
> - **Multimodal Understanding** – integrating Optical, SAR, and IR
> - **Self-Correction** – revising plans based on execution feedback
>
> Closed-form tasks are evaluated programmatically, whereas open-form tasks use structured prompts with **LLM-as-judge** to check both numeric outputs and semantic explanations **(Section 5.2, Figure 4)**.
>
> ---
>
> ### **Addressing Perceived Simplicity in the  Agent Landscape**
>
> GeoHOP is a **benchmark**, not an agent framework.
>
> As shown in **Appendix A.5 Figure 8(Page 9)**, existing remote-sensing benchmarks do not include hierarchical, cognitively grounded geospatial reasoning tasks.
>
> Results in **Appendix A.3 Table 7 (Page 18)** show that even the most advanced RS-MLLMs achieve **very low accuracy** on GeoHOP. This demonstrates that GeoHOP tasks remain highly challenging, despite progress in general-purpose agents.
>
> ### Table 7: Optical accuracy comparison between representative remote-sensing multimodal LLMs (RS-MLLMs) and our **GeoPlanner (GPT-4o)** on GeoHOP optical VQA tasks.
>
> | **Model**                 | **Optical Accuracy (%)** |
> |---------------------------|---------------------------|
> | **RS-MLLMs**              |                           |
> | EarthDial                 | 9.38                      |
> | GeoChat                   | 6.70                      |
> | GeoPix                    | 6.17                      |
> | **LLM-based Agent (Ours)** |                         |
> | **GeoPlanner (GPT-4o)**   | **29.29**                 |
>
>
> ---
>
> ### **Feasibility of Simulation-Based Data Generation for RL**
>
> Although the current pipeline uses real remote-sensing imagery, its scene-context structures and tool-chain abstractions are **programmable**.
> This enables future **simulation-based geospatial environments** for reinforcement learning.
>
> We agree this is a promising direction and plan to explore simulation-driven RL variants of GeoHOP in future work.

---

> ### Author Response · Authors · 2025-11-26
> **Response to Weakness/Questions 2:**
>
> We thank the reviewer for raising the important question of how deep reasoning should be defined and how task difficulty can be meaningfully quantified.
> ### **Distinguishing Deep vs. Shallow Reasoning and Measuring Task Difficulty**
>
> We define **deep reasoning** as hierarchical, cognitively grounded workflows requiring integration of multiple toolkits, rather than executing a single operator.
>
> Task difficulty is quantitatively defined in **Appendix A.1 Table 5(Page 16)** (“Fine-grained results across models and capability subsets”), where tasks are grouped by capability tags:
>
> 1. **S** – Semantic perception
> 2. **G** – Geometric reasoning
> 3. **Q** – Quantitative reasoning
> 4. **H** – Long-horizon planning (≥4 tools)
>
> These tags make the number of reasoning dimensions explicit, providing a transparent and measurable basis for difficulty.
>
> We found:
>
> - API-based models (especially GPT-4o) maintain strong performance across all subsets, including **S+G+Q** and long-horizon **H ∩ (S+G+Q)**.
> - Open-source models often perform well on simpler **S+G** and **S+Q** tasks, but suffer substantial degradation on multi-capability, long-horizon workflows.
>
> This highlights the difficulty of sustained, high-fidelity tool reasoning in extended geospatial pipelines.
> ### **Table 5: Fine-grained results across models and capability subsets**
>
> | Model | Task | Inst. | Tool. | Arg. | Summ. | Ans. |
> |-------|------|-------|-------|------|-------|------|
> | **API-based** |  |  |  |  |  |  |
> | GPT-4o | S+G              | 77.78 | 92.11 | 32.89 | 75.00 | 45.29 |
> |        | S+Q              | 78.33 | 89.20 | 22.80 | 94.00 | 36.00 |
> |        | S+G+Q            | 85.20 | 84.40 | 27.37 | 91.08 | 35.43 |
> |        | H ∩ (S+G+Q)      | 60.70 | 46.77 | 18.09 | 60.47 | 9.81 |
> | Gemini-2.5-Flash | S+G    | 49.44 | 48.03 | 18.42 | 53.57 | 21.43 |
> |        | S+Q              | 30.67 | 25.60 | 12.00 | 30.00 | 8.00 |
> |        | S+G+Q            | 34.51 | 23.40 | 9.00  | 24.93 | 9.97 |
> |        | H ∩ (S+G+Q)      | 50.81 | 34.24 | 12.40 | 41.86 | 0.00 |
> | GPT-3.5 | S+G             | 67.2  | 73.0  | 27.0  | 64.3  | 42.9 |
> |        | S+Q              | 83.3  | 94.4  | 40.8  | 98.0  | 48.0 |
> |        | S+G+Q            | 53.9  | 53.5  | 20.8  | 51.2  | 24.7 |
> |        | H ∩ (S+G+Q)      | 67.1  | 54.4  | 18.3  | 69.8  | 10.5 |
> | **Open-source (Large)** |  |  |  |  |  |  |
> | DeepSeek-V3 | S+G         | 80.0  | 86.8  | 28.3  | 92.9  | 32.1 |
> |        | S+Q              | 83.3  | 96.8  | 30.0  | 98.0  | 40.0 |
> |        | S+G+Q            | 61.9  | 65.7  | 24.0  | 53.8  | 21.3 |
> |        | H ∩ (S+G+Q)      | 70.7  | 54.8  | 17.7  | 60.5  | 5.8 |
> | Qwen2.5-32B-Instruct | S+G | 59.4 | 65.1 | 23.7 | 60.7 | 10.7 |
> |        | S+Q              | 83.3  | 94.4  | 41.2 | 98.0  | 32.0 |
> |        | S+G+Q            | 60.1  | 57.5  | 15.4 | 52.0  | 21.3 |
> |        | H ∩ (S+G+Q)      | 66.0  | 46.5  | 15.2 | 66.3  | 5.8 |
> | Llama-3-70B-Instruct | S+G | 80.00 | 80.26 | 24.34 | 89.29 | 25.00 |
> |        | S+Q              | 83.33 | 92.80 | 38.80 | 94.00 | 34.00 |
> |        | S+G+Q            | 64.44 | 70.12 | 23.17 | 62.99 | 23.10 |
> |        | H ∩ (S+G+Q)      | 72.21 | 47.67 | 15.37 | 80.23 | 11.63 |
> | **Open-source (Medium/Small)** |  |  |  |  |  |  |
> | Qwen2.5-7B-Instruct | S+G  | 80.0  | 94.7  | 32.2  | 85.7  | 42.9 |
> |        | S+Q              | 83.3  | 96.0  | 32.4  | 100.0 | 34.0 |
> |        | S+G+Q            | 78.7  | 77.7  | 23.5  | 57.5  | 22.0 |
> |        | H ∩ (S+G+Q)      | 69.0  | 53.6  | 17.1  | 73.3  | 12.8 |
> | InternLM3-8B-Instruct | S+G | 74.4 | 82.9 | 28.9 | 78.6 | 25.0 |
> |        | S+Q              | 83.3  | 94.4  | 41.2 | 100.0 | 32.0 |
> |        | S+G+Q            | 55.8  | 55.2  | 22.2 | 38.6  | 19.7 |
> |        | H ∩ (S+G+Q)      | 70.2  | 49.6  | 16.8 | 64.0  | 5.8 |
> | LLaMA3-1-8B | S+G         | 80.0  | 89.5  | 28.3  | 85.7  | 21.4 |
> |        | S+Q              | 51.7  | 51.6  | 28.4 | 40.0  | 6.0 |
> |        | S+G+Q            | 71.2  | 71.0  | 23.2 | 50.7  | 19.2 |
> |        | H ∩ (S+G+Q)      | 67.3  | 48.7  | 16.0 | 76.7  | 10.5 |
> | Phi-3-Mini-4K-Instruct | S+G | 51.11 | 42.11 | 14.47 | 32.14 | 7.14 |
> |        | S+Q              | 77.67 | 82.00 | 32.40 | 72.00 | 16.00 |
> |        | S+G+Q            | 57.35 | 40.98 | 18.34 | 2.62  | 2.62 |
> |        | H ∩ (S+G+Q)      | 41.86 | 28.29 | 8.79  | 0.00  | 0.00 |
> | Mistral-7B-Instruct-v0.2 | S+G | 73.9 | 74.3 | 24.3 | 78.6 | 25.0 |
> |        | S+Q              | 77.7  | 86.0  | 36.4 | 92.0  | 16.0 |
> |        | S+G+Q            | 56.0  | 43.6  | 15.1 | 37.3  | 13.6 |
> |        | H ∩ (S+G+Q)      | 65.3  | 40.8  | 14.3 | 47.7  | 5.8 |
> | Yi-1.5-6B-Chat | S+G      | 81.67 | 63.82 | 23.68 | 85.71 | 14.29 |
> |        | S+Q              | 54.33 | 56.40 | 21.20 | 38.00 | 14.00 |
> |        | S+G+Q            | 60.37 | 41.02 | 15.49 | 30.97 | 11.02 |
> |        | H ∩ (S+G+Q)      | 73.37 | 36.82 | 14.60 | 53.49 | 6.98 |
>
> ---

---

> > ### Author Response · Authors · 2025-11-26
> > **Response to Weakness/Questions 3:**
> >
> > We thank the reviewer for highlighting the need to clarify how agent capabilities are measured.
> >
> > # Measuring Agent Capabilities Across Dimensions
> >
> > We have clarified **GeoPlanner’s behavior** in the revised paper:
> >
> > ## **Main results — Section 5.2 (Table 3, Page 8)**
> > ### Table 3: Main results of GeoPlanner on the GeoHOP benchmark
> >
> > |Model|Inst.|Tool.|Arg.|Summ.|P|O|L|Ans.|
> > |-|-|-|-|-|-|-|-|-|
> > |**API-based**|||||||||
> > |GPT-4o|**_79.40_**|72.01|**_23.57_**|**_82.83_**|**_72.73_**|**_88.38_**|77.95|**_25.70_**|
> > |Gemini-2.5-Flash|59.09|43.31|15.75|47.22|41.47|40.89|41.51|12.04|
> > |GPT-3.5|59.05|57.12|21.88|59.08|61.22|61.04|53.30|24.40|
> > |**Open-source (Large)**|||||||||
> > |DeepSeek-V3|65.84|66.33|23.32|60.92|65.44|66.04|57.93|21.28|
> > |Qwen2.5-32B-Instruct|62.83|58.00|17.41|58.90|44.28|67.40|61.47|19.08|
> > |Llama-3-70B-Instruct|67.89|67.47|22.66|69.91|62.72|71.62|62.54|22.75|
> > |**Open-source (Medium/Small)**|||||||||
> > |Qwen2.5-7B-Instruct|77.19|**_74.71_**|23.11|65.32|68.56|73.03|**_82.42_**|22.57|
> > |InternLM3-8B-Instruct|61.26|57.72|22.59|50.28|61.44|53.70|60.70|18.90|
> > |LLaMA3-1-8B|69.46|65.99|22.27|55.60|68.42|60.34|70.82|16.70|
> > |Phi-3-Mini-4K-Instruct|55.45|41.14|17.17|10.09|43.63|7.52|51.74|3.67|
> > |Mistral-7B-Instruct-v0.2|60.03|47.05|16.73|46.06|49.75|44.05|45.23|13.21|
> > |Yi-1.5-6B-Chat|63.38|42.08|16.00|37.98|47.44|38.73|36.02|10.83|
> >
> >
> > ---
> >
> > ### Notes
> >
> > - **Inst., Tool., Arg., Summ.** denote *InstAcc*, *ToolAcc*, *ArgAcc*, *SummAcc* (step-by-step metrics).
> > - **P, O, L** denote end-to-end scores for **Perception**, **Spatial Statistics**, and **Spatial Relations** toolkits.
> > - **Ans.** = final answer accuracy (**AnsAcc**).
> > - **Bold** = best score across all models.
> > - _Underline_ = best within the same model scale.
> > Reporting step-by-step metrics of instruction-following, tool selection, argument correctness, and summary generation, respectively, and end-to-end scores (P, O, and L) within the Perception, Spatial-Statistics, and Spatial-Relations toolkits.
> >
> > Across all models, **argument consistency (ArgAcc) is the primary bottleneck**, and **poor tool selection (ToolAcc) strongly correlates with reduced end-to-end accuracy**. Even the strongest systems achieve only modest absolute scores, underscoring the difficulty of reliable geospatial reasoning, accurate argument propagation, and faithful summary generation in GeoHOP.
> >
> > ---
> >
> > ###  **Error analysis — Section 5.4 (Table 4, Page 10)**
> > ### **Table 4: Percentage distribution of four categories of errors for different LLMs**
> >
> > |Model|FmtErr(%)|ReasonErr(%)|PercErr(%)|ToolErr(%)|
> > |-|-|-|-|-|
> > |**GPT-4o**|41.65|13.49|41.26|3.60|
> > |**Gemini-2.5-Flash**|29.47|21.18|12.09|37.26|
> > |**GPT-3.5**|31.20|18.60|10.70|39.50|
> > |**DeepSeek-V3**|66.40|23.00|8.70|1.90|
> > |**Qwen2.5-32B-Instruct**|31.27|19.78|13.99|34.96|
> > |**Llama-3-70B-Instruct**|55.20|25.70|18.20|0.90|
> > |**Qwen2.5-7B-Instruct**|42.65|23.48|29.37|4.50|
> > |**InternLM3-8B-Instruct**|32.70|23.90|11.40|32.00|
> > |**LLaMA3-1-8B**|38.36|24.58|20.28|16.78|
> > |**Phi-3-Mini-4K-Instruct**|45.60|34.70|1.80|17.90|
> > |**Mistral-7B-Instruct-v0.2**|45.40|21.20|5.40|28.00|
> > |**Yi-1.5-6B-Chat**|37.90|28.60|8.60|24.90|
> >
> > Further breaks down failures into **format, reasoning, perception, and tool-use errors**, enabling diagnosis across cognitive dimensions.
> >
> > Results indicate that **robust structured output generation and stable multimodal perceptual grounding—not abstract reasoning—are the key limitations** for current geospatial agents, highlighting the need for better format alignment and stronger cross-modal grounding in future systems.
> >
> > ---
> >
> > ### **Modality-specific performance differences in Appendix A.2 (Table 6)**
> > ### **Table 6: Accuracy comparison across models on the overall benchmark and each sensing modality**
> >
> > |Model|Overall|Opt.|SAR|IR|
> > |-|-|-|-|-|
> > |**GPT-4o**|**25.70**|**29.29**|11.50|**100.00**|
> > |GPT-3.5|24.40|28.05|0.00|56.82|
> > |Gemini-2.5-Flash|12.04|11.63|0.00|**100.00**|
> > |DeepSeek-V3|21.28|25.19|0.00|43.18|
> > |Qwen2.5-32B-Instruct|19.08|23.12|0.00|34.09|
> > |Qwen2.5-7B-Instruct|22.57|22.08|17.24|40.91|
> > |Llama-3.1-8B-Instruct|16.70|14.81|**26.72**|6.82|
> > |Llama-3-70B-Instruct|22.75|27.27|0.00|43.18|
> > |InternLM3-8B-Instruct|18.90|22.08|0.86|38.64|
> > |Mistral-7B-Instruct-v0.2|13.21|16.36|0.00|20.45|
> > |Phi-3-Mini-4K-Instruct|3.67|0.52|8.62|18.18|
> > |Yi-1.5-6B-Chat|10.83|13.51|0.00|15.91|
> >
> > Now reports **modality-dependent accuracy**, showing expected differences across Optical / SAR / IR:
> >
> > - **Optical:** moderate accuracy for high-capacity controllers.
> > - **SAR:** largest drop due to texture/speckle noise; many open-source models perform poorly.
> > - **IR:** frontier models achieve near-perfect scores; smaller models remain below 45%.
> >
> > GeoPlanner requires **no modality-specific tuning**: tool retrieval is semantic and modality-tagged, with automatic selection of compatible operators.

---

> > > ### Comment · Reviewer_CA8L · 2025-11-28
> > >
> > > We sincerely thank the authors for their tremendous effort; the quality of the manuscript has indeed improved substantially. This is an important and valuable line of work. However, I still strongly encourage the authors to engage more deeply with the broader Agent literature beyond the remote sensing (RS) community; otherwise, the paper will struggle to provide genuinely new insights.
> > >
> > > At present (in Table 7.), the design of Agents in RS lags significantly behind what is being done in industry. I suggest the authors conduct a small exploratory experiment: for example, integrating DINO-X via MCP into Cursor/Trae for VQA. I believe that, after running such tests, the authors will better understand the motivation behind my comments.
> > >
> > > If one adopts a combination of a code-based Agent and MCP, the evaluation framework proposed in this paper cannot be fully applied to such a design—yet this is precisely the direction in which future RS Agent designs are likely to evolve.
> > >
> > > In summary, my score lies between 4 and 6, depending on whether the authors have a clear plan for the subsequent improvements along these lines.

---

> > > > ### Author Response · Authors · 2025-12-02
> > > > **Response to exploratory experiment:**
> > > >
> > > > We sincerely thank the reviewer for the constructive and insightful feedback, especially the recommendation to conduct a **“small exploratory experiment”** comparing our **tool-augmented agent** with code-based industrial paradigms. Following this guidance, we evaluated a **Code-based Agent** (Cursor/MCP + RemoteSAM) against our **Tool-augmented Agent (GeoPlanner)** on detection tasks.
> > > >
> > > > As shown in **Appendix A.7 (Table 8)(Page 25)**, the results present a more nuanced picture than final accuracy alone. While the code-based agent achieves higher answer accuracy, our tool-augmented agent demonstrates substantially stronger operational stability and enables fine-grained, interpretable analyses, which are essential for controllability in RS applications.
> > > >
> > > > The exploratory experiment highlights both the potential and the practical challenges of adopting code-based architectures in RS. We fully agree that integrating the strengths of both paradigms represents a key direction for the future evolution of RS agents.
> > > >
> > > > **Table 8:** Comparison of Code-based Agent (CBA, i.e., Cursor) and Tool-Augmented Agent (TAA) on the Detection Task Subset. ($\mathcal{O}$ represents the accuracy of spatial statistics tool calls; $\mathcal{L}$ represents the accuracy of spatial relations tool calls; and Ans. represents the final answer accuracy.)
> > > >
> > > > | Agent Type | $\mathcal{O}$ (Operation) | $\mathcal{L}$ (Logic) | Ans. (Accuracy) |
> > > > | :--- | :---: | :---: | :---: |
> > > > | **Ours (Tool-Augmented Agent)** | **99.23%** | **86.51%** | 40.11% |
> > > > | **Cursor (Code-based Agent)** | 56.50% | 85.88% | **44.63%** |

---

### Official Review · Reviewer_guxm · 2025-10-31

**Soundness:** 3
**Presentation:** 3
**Contribution:** 3
**Rating:** 6
**Confidence:** 3

**Summary:**

This paper has two main contributions: First the authors introduce a benchmark (GeoHOP) of tasks for geospatial reasoning, including 417 tasks. The tasks require combining different elements of geospatial reasoning, with three elements: spatial trigger, expert interpretation and tool use. Secondly the authors introduce an agent "Geoplanner" to solve these tasks and present performance metrics for several frontier models using the agent. The planner is given different sets of tools (open source models), that are chosen by a semantic retriever to solve the tasks. The planner makes a plan, tries to execute the plan, and then retries if there are errors. There is an overall error budget to control cost. The agent is evaluated on Answer accuracy and argument consistency, evaluated by an LLM as a judge

**Strengths:**

The benchmark seems creative though this is hard for me to judge as I only can see the examples in the paper.  The geoplanner is quite sensible.

**Weaknesses:**

I can't tell from the paper how good this benchmark is. The examples that are given don't show the potential breadth of what is in the benchmark, and its utility depends critically on the level of difficulty and relevance of the questions. The low scores of the fronteir models on the tasks make it seem that the benchmark is hard -- but i don't have a sense for _why_ it is hard, and what are the types of mistakes. The car counting example in the paper is quite nice and clear, but it would be easier to assess the importance of this benchmark if I could understand its breadth better.

At the same time, I'm not a great fan of the LLM as a judge evaluation of the agents. For the car example the answer is the number of cars. Can't this be evaluated programatically? I'd much prefer if the evals were as rigorous as possible and i think LLM as a judge weakens the argument

I'd like to understand whether the tools were chosen specifically to improve performance on the benchmark. Please give justification for the tools that you hvae chosen and explain whether or not they were chosen independently of the questions in the benchmark. If there was dependence please explain how you know you aren't cherry picking. At an extreme limit we build a model for every problem and simply call the model as a tool, so the agent only has to distinguish which model it is choosing. This is not interesting.

**Questions:**

1. Clearer discussion of the types and variability of problems in the benchmark. What does it cover, what does it not cover, what is the level of difficulty.  The current discussion is opaque. This will be easier I guess when we can see the benchmark but still a summary in the paper is good.
1'. Comparison to other benchmarks -- what is new here.
2. More information about the actions the geoplanner takes and the distribution of errors across the tasks. I don't understand the red x's and checkmarks in figure 1. The caption is rather brief. how much of it has to do with ability to use the particular tools you are letting it call.
2'. How does geoplanner do on other benchmarks where this is relevant. or if it isn't relevant because the geoplanner is very specific in the tools it can use then it would be good to understand why. this coudl be a major limitation of the study depending on the answer.
3. Explain how the tools were chosen and whether or not you were cherry picking tools to problems like i outlined above. Measuring performance on another benchmark (eg https://openreview.net/pdf?id=oaYShIy3Xe) would help sort this out, as one woudl expect lower performances if the tools were cherry picked

---

> ### Author Response · Authors · 2025-11-26
> **Response to Weakness/Questions 1:**
>
> We thank the reviewer for outlining these weaknesses and questions, which strengthened the clarity and rigor of our paper.
> ## **Benchmark Scope, Diversity, and Difficulty**
>
> To clarify breadth and difficulty, we added two supplements:
>
> - **Appendix A.6(Page 20)**: a structured Scene-Context Knowledge Base covering all four domain-knowledge families, with geometric/relational elements and representative example queries.
> - **Appendix A.5 (Figure 8)(Page 20)**: a Benchmark Comparison Overview, placing GeoHOP examples alongside those from existing RS benchmarks, making scope and difficulty directly comparable.
>
> These additions summarize what the benchmark covers, what is out-of-scope, and why tasks are challenging.
>
> ---
>
> ## **Evaluation Methodology (LLM-as-Judge)**
>
> We agree that programmatic evaluation is preferable when applicable. However, many GeoHOP tasks require **joint correctness of a numeric value and a scene-based explanation**. As shown in the new example in **Figure 4 (Section 5.2)(Page 9)**, correctness cannot be determined by counting objects alone. In such cases, an LLM-as-judge is necessary to evaluate semantic and explanatory components in a structured, reference-grounded manner.
>
> ---
>
> ## **Tool Usage and Capabilities**
>
> We have clarified GeoPlanner’s behavior in the revised paper:
>
> - The **Figure 1 caption(Page 4)** now explicitly explains the **red X (incorrect)** and **green checkmark (validated)** symbols.
>   Red crosses mark erroneous intermediate plans or tool calls (e.g., reasoning, selection, ordering, argument, or compatibility errors), while green checkmarks indicate the validated final tool chain and answer.
>
> - **Section 5.2 (Table 3)(Page 8)** reports step-by-step metrics of instruction-following, tool selection, argument correctness, and summary generation, as well as end-to-end scores (P, O, and L) within the Perception, Spatial-Statistics, and Spatial-Relations toolkits.
>
> ### Table 3: Main results of GeoPlanner on the GeoHOP benchmark
>
> | Model | Inst. | Tool. | Arg. | Summ. | P | O | L | Ans. |
> |-------|-------|-------|-------|-------|-------|-------|-------|-------|
> | **API-based** |||||||||
> | GPT-4o | **_79.40_** | 72.01 | **_23.57_** | **_82.83_** | **_72.73_** | **_88.38_** | 77.95 | **_25.70_** |
> | Gemini-2.5-Flash | 59.09 | 43.31 | 15.75 | 47.22 | 41.47 | 40.89 | 41.51 | 12.04 |
> | GPT-3.5 | 59.05 | 57.12 | 21.88 | 59.08 | 61.22 | 61.04 | 53.30 | 24.40 |
> | **Open-source (Large)** |||||||||
> | DeepSeek-V3 | 65.84 | 66.33 | 23.32 | 60.92 | 65.44 | 66.04 | 57.93 | 21.28 |
> | Qwen2.5-32B-Instruct | 62.83 | 58.00 | 17.41 | 58.90 | 44.28 | 67.40 | 61.47 | 19.08 |
> | Llama-3-70B-Instruct | 67.89 | 67.47 | 22.66 | 69.91 | 62.72 | 71.62 | 62.54 | 22.75 |
> | **Open-source (Medium/Small)** |||||||||
> | Qwen2.5-7B-Instruct | 77.19 | **_74.71_** | 23.11 | 65.32 | 68.56 | 73.03 | **_82.42_** | 22.57 |
> | InternLM3-8B-Instruct | 61.26 | 57.72 | 22.59 | 50.28 | 61.44 | 53.70 | 60.70 | 18.90 |
> | LLaMA3-1-8B | 69.46 | 65.99 | 22.27 | 55.60 | 68.42 | 60.34 | 70.82 | 16.70 |
> | Phi-3-Mini-4K-Instruct | 55.45 | 41.14 | 17.17 | 10.09 | 43.63 | 7.52 | 51.74 | 3.67 |
> | Mistral-7B-Instruct-v0.2 | 60.03 | 47.05 | 16.73 | 46.06 | 49.75 | 44.05 | 45.23 | 13.21 |
> | Yi-1.5-6B-Chat | 63.38 | 42.08 | 16.00 | 37.98 | 47.44 | 38.73 | 36.02 | 10.83 |
>
> ---
>
> ### Notes
>
> - **Inst., Tool., Arg., Summ.** denote *InstAcc*, *ToolAcc*, *ArgAcc*, *SummAcc* (step-by-step metrics).
> - **P, O, L** denote end-to-end scores for **Perception**, **Spatial Statistics**, and **Spatial Relations** toolkits.
> - **Ans.** = final answer accuracy (**AnsAcc**).
> - **Bold** = best score across all models.
> - _Underline_ = best within the same model scale.
>
> ---
> ## **Tool Selection and “Cherry-Picking” Concerns**
>
> Tools were not chosen to artificially boost performance. As stated in **Section 4.1(Page 6)**, the tool library is fixed to five standard geospatial operator families common to RS workflows. All GeoHOP tasks were constructed in **Section 3.2(Page 3)** through domain knowledge and expert review before tool selection or implementation, ensuring independence between task design and tool inventory.
>
> ---
>
> ## **Evaluation on Other Benchmarks (Fairness & Generality)**
>
> We acknowledge the need to evaluate GeoPlanner beyond GeoHOP. While full cross-benchmark studies are planned, GeoPlanner’s rich, extensible toolset—covering perception, spatial relations, spatial statistics, and temporal/change detection across Optical, SAR, and IR—reflects **domain-general operations** in remote sensing, not GeoHOP-specific designs. This allows direct adaptation to other RS benchmarks by mapping their requirements to existing or easily extendable tools, enabling broader comparisons and transfer to diverse scenarios.

---

> ### Author Response · Authors · 2025-11-26
> **Response to Weakness/Questions 2:**
>
> We thank the reviewer for noting the importance of detailing the error types.
> ## **Fine-grained Error Analysis**
>
> In the revised version, we also added a fine-grained error breakdown (**Section 5.4, Table 4 (Page 10)**), covering:
>
> 1. **Format errors** — invalid/incomplete JSON or argument fields.
> 2. **Reasoning errors** — logical mistakes with correct tool/perception input.
> 3. **Perception errors** — misinterpretation of visual/geospatial inputs, including masks, change maps, and spatial layouts.
> 4. **Tool-use errors** — incorrect operator choice, missing steps, or invalid parameters (e.g., incorrect buffer distances).
>
> Results indicate that **robust structured output generation** and **stable multimodal perceptual grounding**—not abstract reasoning—are the key limitations for current geospatial agents, highlighting the need for better format alignment and stronger cross-modal grounding in future systems.
> ### **Table 4: Percentage distribution of four categories of errors for different LLMs**
>
> | **Model** | **Format Error (%)** | **Reasoning Error (%)** | **Perception Error (%)** | **Tool-Use Error (%)** |
> |-----------|----------------------|---------------------------|----------------------------|--------------------------|
> | **GPT-4o** | 41.65 | 13.49 | 41.26 | 3.60 |
> | **Gemini-2.5-Flash** | 29.47 | 21.18 | 12.09 | 37.26 |
> | **GPT-3.5** | 31.20 | 18.60 | 10.70 | 39.50 |
> | **DeepSeek-V3** | 66.40 | 23.00 | 8.70 | 1.90 |
> | **Qwen2.5-32B-Instruct** | 31.27 | 19.78 | 13.99 | 34.96 |
> | **Llama-3-70B-Instruct** | 55.20 | 25.70 | 18.20 | 0.90 |
> | **Qwen2.5-7B-Instruct** | 42.65 | 23.48 | 29.37 | 4.50 |
> | **InternLM3-8B-Instruct** | 32.70 | 23.90 | 11.40 | 32.00 |
> | **LLaMA3-1-8B** | 38.36 | 24.58 | 20.28 | 16.78 |
> | **Phi-3-Mini-4K-Instruct** | 45.60 | 34.70 | 1.80 | 17.90 |
> | **Mistral-7B-Instruct-v0.2** | 45.40 | 21.20 | 5.40 | 28.00 |
> | **Yi-1.5-6B-Chat** | 37.90 | 28.60 | 8.60 | 24.90 |

---

### Official Review · Reviewer_PTWu · 2025-10-31

**Soundness:** 2
**Presentation:** 2
**Contribution:** 2
**Rating:** 4
**Confidence:** 5

**Summary:**

The paper presents GeoHOP, a new benchmark designed to assess hierarchical geospatial reasoning through 417 curated, scenario-based tasks spanning optical, SAR, and IR imagery. These tasks represent applied domains such as urban heat and hazard vulnerability assessment, etc., requiring multi-step reasoning that combines perception with operators such as buffer, overlap, and others. To address these tasks, the authors introduce GeoPlanner, an LLM-based agent framework that organizes geospatial tools into a hierarchical library. The agent retrieves relevant tools, generates modality-aware reasoning plans, executes them, and composes final responses grounded in tool-derived outputs. The study evaluates some large language models, including GPT-4o, Claude-4, and Qwen-2.5, on GeoHOP using structured, step-wise metrics. GPT-4o achieves the strongest overall performance. The dataset is constructed through a two-stage curation pipeline, combining knowledge-augmented generation with expert adjudication.

**Strengths:**

This work targets geospatial reasoning, offering 417 scenario-driven, hierarchy-aware tasks that span optical, SAR, and IR, useful coverage for multimodal evaluation. Its benchmark instances are structured for verifiability via expert interpretation and a verifiable tool chain, which bridges perception to higher-level analysis. GeoPlanner agent adds practical value with a hierarchical, typed tool library, modality awareness, and fault-tolerant execution, improving robustness for long-horizon reasoning. The dataset curation is documented through a two-stage pipeline with expert adjudication. The paper also provides step-wise and end-to-end evaluation across several LLMs, establishing initial baselines.

**Weaknesses:**

**Small scale with modality imbalance.** The dataset includes 417 total tasks, 252 using optical images, 119 using SAR, and only 42 using infrared data. Overall, the benchmark is relatively small and not well-balanced across modalities, which limits its ability to represent the full diversity of real-world remote-sensing scenarios. The small size reduces confidence that model performance will generalize beyond the sampled tasks.

**No explicit temporal/change tasks.** All examples are single snapshots; there is no bi-temporal change detection or time-aware operator, even though change analysis is central to many remote-sensing applications. ThinkGeo(Shabbir et al., 2025) includes explicit temporal-change tasks.

**Human verification detail.** The authors mention expert adjudication by remote sensing specialists, but the paper does not quantify human-hour effort, inter-annotator agreement, or error rates. This makes it difficult to assess the depth and reliability of the quality control.

**Latency/throughput not reported.** The paper analyzes success and failure counts and step lengths but does not provide runtime or throughput measurements (step-wise vs. end-to-end). For practical benchmarking, such efficiency metrics would be valuable.

**Limited insight by category/model.** Several LLMs (GPT family, Claude-4, Gemini-2.5, DeepSeek-V3, Qwen-2.5) are benchmarked, yet the analysis remains coarse-grained. There is no breakdown showing which models perform better for specific task types or modalities, limiting interpretability and future insights.

**Geographic coverage.** The dataset sources (LoveDA, Potsdam, OGSOD, DMIST) are listed, but the paper does not describe the global or regional distribution of scenarios. Without geographic context, it is unclear how representative or generalizable the benchmark is.

**Tools are narrower.** The toolkit focuses on perception and spatial-relation operators (buffer, overlap, containment) but lacks temporal/change and code-based tools that are common in broader geospatial reasoning workflows. This constrains its applicability to real-world analytical pipelines.

**Questions:**

I would appreciate additional clarification on several aspects of the work. Could the authors elaborate on how the benchmark's relatively small and unbalanced scale across modalities (Optical, SAR, IR) affects its representativeness and generalization potential? It would also be helpful to understand why temporal or change-detection tasks were not included, given their central importance in remote-sensing reasoning. Further details on the human verification process, such as total annotation effort, inter-annotator agreement, and error rates, would strengthen confidence in dataset quality. Reporting runtime or throughput metrics, along with a breakdown of model performance across task categories or modalities, would also improve interpretability. Finally, clarification on the scope of tool diversity would help assess practical relevance. Please see the weaknesses for details.

---

> ### Author Response · Authors · 2025-11-26
> **Response to Weakness/Questions 1:**
>
> We thank the reviewer for the detailed and constructive comments.
>
>  **Dataset scale**
>
> In the revised manuscript, **Section 3.2, Table 1(Page 5)** expands the benchmark from **417 to 545** instances by adding **3 new datasets**, including **2 city-diversified land cover datasets** and a **semantic change-detection dataset**, which directly increases **multi-temporal** and **scene diversity**. While the modalities remain numerically imbalanced—reflecting real-world practice where optical data are far more prevalent—we ensure that SAR and IR samples come from multiple geographically diverse regions.
>
> We also add **Appendix A.6 (Scene-Context Knowledge Base) (Page 20)**, which lists all domain-grounded scene types and example queries, demonstrating that GeoHOP captures substantially richer spatial complexity. This includes varied land-cover categories, spatial configurations, and operational contexts relevant to real-world geospatial reasoning.
> ### **Table 1: Datasets used as image sources in the GeoHOP benchmark**
>
> | **Name** | **Annotation Type** | **Resolution** | **Modality** | **Geographical Coverage** |
> |---------|----------------------|----------------|--------------|----------------------------|
> | LoveDA | Masks | 0.3 m/px | Optical | **Nanjing / Changzhou / Wuhan, China** |
> | ISPRS Potsdam | Masks | 0.05 m/px | Optical | **Potsdam, Germany** |
> | **ISPRS Vaihingen** | Masks | 0.09 m/px | Optical | **Vaihingen, Germany** |
> | **DeepGlobe Land Cover** | Masks | 0.50 m/px | Optical | **Cities in India / Indonesia / Thailand** |
> | **HRSCD** | Change masks | 0.45 m/px | Optical | **Rennes / Caen, France** |
> | OGSOD | Bounding boxes | 3 m/px | SAR | **Cities in China** |
> | DMIST | Bounding boxes | -- | IR | **Chengdu, China** |
>
>
> ---
>
> **Temporal/Change-Detection Tasks**
>
> We have newly incorporated an **optical change-detection module** into the Perception toolkit, in order to support **bi-temporal spatial reasoning (Section 4.1) (Page 6)**. GeoHOP now includes explicit bi-temporal tasks enabled by these additions, such as detecting land-cover change within specified buffers and reasoning over time-varying spatial patterns.
>
> ---
>
>  **Human Verification**
>
> As described in **Section 3.2 (Page 3)**, all instances undergo a **three-pass hierarchical review** by remote-sensing experts using a standardized rubric. The process checks semantic integrity, tool-sequence coherence, and geometric correctness, with discrepancies resolved through discussion before acceptance. This rigorous validation ensures high quality and domain accuracy across all modalities and task types.
>
> ---
>
> **Runtime and Throughput Analysis**
>
> We have added a computational cost analysis in **Appendix A.4 (Figure 6 and Figure 7)(Page 19)**, including:
>
> 1. Per-task inference latency.
> 2. Total end-to-end runtime for the 545-task benchmark.
> 3. Tool-usage breakdown per model (LLM reasoning vs. perception vs. spatial-analysis tools).
>
> These additions allow readers to assess both accuracy and efficiency trade-offs.
> ### **Fine-Grained Evaluation Across Modalities**
>
> The revised **Appendix A.2 (Table 6)(Page 17)** now reports modality-dependent accuracy, showing expected differences across Optical/SAR/IR:
>
>   (1) **Optical:** moderate accuracy for high-capacity controllers.
>   (2) **SAR:** largest drop due to texture/speckle noise; many open-source models perform poorly.
>   (3) **IR:** frontier models achieve near-perfect scores, smaller models remain below 45%.
>
> GeoPlanner requires no modality-specific tuning: tool retrieval is semantic and modality-tagged, with automatic selection of compatible operators.
> ### **Table 6: Accuracy comparison across models on the overall benchmark and each sensing modality**
>
> | **Model** | **Overall** | **Optical** | **SAR** | **IR** |
> |-----------|-------------|-------------|---------|--------|
> | **GPT-4o** | **25.70** | **29.29** | 11.50 | **100.00** |
> | GPT-3.5 | 24.40 | 28.05 | 0.00 | 56.82 |
> | Gemini-2.5-Flash | 12.04 | 11.63 | 0.00 | **100.00** |
> | DeepSeek-V3 | 21.28 | 25.19 | 0.00 | 43.18 |
> | Qwen2.5-32B-Instruct | 19.08 | 23.12 | 0.00 | 34.09 |
> | Qwen2.5-7B-Instruct | 22.57 | 22.08 | 17.24 | 40.91 |
> | Llama-3.1-8B-Instruct | 16.70 | 14.81 | **26.72** | 6.82 |
> | Llama-3-70B-Instruct | 22.75 | 27.27 | 0.00 | 43.18 |
> | InternLM3-8B-Instruct | 18.90 | 22.08 | 0.86 | 38.64 |
> | Mistral-7B-Instruct-v0.2 | 13.21 | 16.36 | 0.00 | 20.45 |
> | Phi-3-Mini-4K-Instruct | 3.67 | 0.52 | 8.62 | 18.18 |
> | Yi-1.5-6B-Chat | 10.83 | 13.51 | 0.00 | 15.91 |
> ---
> ### **Geographic Coverage**
>
> To address representativeness concerns, we updated **Section 3, Table 1(Page 5)** with a “Geographical Coverage” column listing the specific cities and countries represented in each source dataset. This clarifies that GeoHOP spans diverse urban, rural, coastal, and agricultural regions across multiple countries and climate zones.

---

> > ### Author Response · Authors · 2025-11-26
> > **Response to Weakness/Questions 2:**
> >
> > We thank the reviewer for highlighting the need for category-level analysis and narrower tools.
> > ### **Fine-Grained Evaluation Across Task Types**
> >
> > To make GeoHOP’s evaluable capabilities explicit, we introduced capability tags for:
> >
> >   (1) **S:** Semantic perception.
> >       **(2) G:** Geometric reasoning.
> >       **(3) Q:** Quantitative reasoning.
> >       **(4) H:** Long-horizon planning (≥4 tools).
> >
> > We present this in **Appendix A.1, Table 5(Page 16)** (“Fine-grained results across models and capability subsets”).
> > API-based models, especially GPT-4o, maintain strong performance across all subsets, particularly complex integrations like **S+G+Q** and long-horizon **H ∩ (S+G+Q)**. Open-source models often perform well on simpler **S+G** and **S+Q** tasks, but suffer substantial degradation on multi-capability, long-horizon workflows—underscoring the difficulty of sustained, high-fidelity tool reasoning in extended geospatial pipelines.
> > ### **Table 5: Fine-grained results across models and capability subsets**
> >
> > | Model | Task | Inst. | Tool. | Arg. | Summ. | Ans. |
> > |-------|------|-------|-------|------|-------|------|
> > | **API-based** |  |  |  |  |  |  |
> > | GPT-4o | S+G              | 77.78 | 92.11 | 32.89 | 75.00 | 45.29 |
> > |        | S+Q              | 78.33 | 89.20 | 22.80 | 94.00 | 36.00 |
> > |        | S+G+Q            | 85.20 | 84.40 | 27.37 | 91.08 | 35.43 |
> > |        | H ∩ (S+G+Q)      | 60.70 | 46.77 | 18.09 | 60.47 | 9.81 |
> > | Gemini-2.5-Flash | S+G    | 49.44 | 48.03 | 18.42 | 53.57 | 21.43 |
> > |        | S+Q              | 30.67 | 25.60 | 12.00 | 30.00 | 8.00 |
> > |        | S+G+Q            | 34.51 | 23.40 | 9.00  | 24.93 | 9.97 |
> > |        | H ∩ (S+G+Q)      | 50.81 | 34.24 | 12.40 | 41.86 | 0.00 |
> > | GPT-3.5 | S+G             | 67.2  | 73.0  | 27.0  | 64.3  | 42.9 |
> > |        | S+Q              | 83.3  | 94.4  | 40.8  | 98.0  | 48.0 |
> > |        | S+G+Q            | 53.9  | 53.5  | 20.8  | 51.2  | 24.7 |
> > |        | H ∩ (S+G+Q)      | 67.1  | 54.4  | 18.3  | 69.8  | 10.5 |
> > | **Open-source (Large)** |  |  |  |  |  |  |
> > | DeepSeek-V3 | S+G         | 80.0  | 86.8  | 28.3  | 92.9  | 32.1 |
> > |        | S+Q              | 83.3  | 96.8  | 30.0  | 98.0  | 40.0 |
> > |        | S+G+Q            | 61.9  | 65.7  | 24.0  | 53.8  | 21.3 |
> > |        | H ∩ (S+G+Q)      | 70.7  | 54.8  | 17.7  | 60.5  | 5.8 |
> > | Qwen2.5-32B-Instruct | S+G | 59.4 | 65.1 | 23.7 | 60.7 | 10.7 |
> > |        | S+Q              | 83.3  | 94.4  | 41.2 | 98.0  | 32.0 |
> > |        | S+G+Q            | 60.1  | 57.5  | 15.4 | 52.0  | 21.3 |
> > |        | H ∩ (S+G+Q)      | 66.0  | 46.5  | 15.2 | 66.3  | 5.8 |
> > | Llama-3-70B-Instruct | S+G | 80.00 | 80.26 | 24.34 | 89.29 | 25.00 |
> > |        | S+Q              | 83.33 | 92.80 | 38.80 | 94.00 | 34.00 |
> > |        | S+G+Q            | 64.44 | 70.12 | 23.17 | 62.99 | 23.10 |
> > |        | H ∩ (S+G+Q)      | 72.21 | 47.67 | 15.37 | 80.23 | 11.63 |
> > | **Open-source (Medium/Small)** |  |  |  |  |  |  |
> > | Qwen2.5-7B-Instruct | S+G  | 80.0  | 94.7  | 32.2  | 85.7  | 42.9 |
> > |        | S+Q              | 83.3  | 96.0  | 32.4  | 100.0 | 34.0 |
> > |        | S+G+Q            | 78.7  | 77.7  | 23.5  | 57.5  | 22.0 |
> > |        | H ∩ (S+G+Q)      | 69.0  | 53.6  | 17.1  | 73.3  | 12.8 |
> > | InternLM3-8B-Instruct | S+G | 74.4 | 82.9 | 28.9 | 78.6 | 25.0 |
> > |        | S+Q              | 83.3  | 94.4  | 41.2 | 100.0 | 32.0 |
> > |        | S+G+Q            | 55.8  | 55.2  | 22.2 | 38.6  | 19.7 |
> > |        | H ∩ (S+G+Q)      | 70.2  | 49.6  | 16.8 | 64.0  | 5.8 |
> > | LLaMA3-1-8B | S+G         | 80.0  | 89.5  | 28.3  | 85.7  | 21.4 |
> > |        | S+Q              | 51.7  | 51.6  | 28.4 | 40.0  | 6.0 |
> > |        | S+G+Q            | 71.2  | 71.0  | 23.2 | 50.7  | 19.2 |
> > |        | H ∩ (S+G+Q)      | 67.3  | 48.7  | 16.0 | 76.7  | 10.5 |
> > | Phi-3-Mini-4K-Instruct | S+G | 51.11 | 42.11 | 14.47 | 32.14 | 7.14 |
> > |        | S+Q              | 77.67 | 82.00 | 32.40 | 72.00 | 16.00 |
> > |        | S+G+Q            | 57.35 | 40.98 | 18.34 | 2.62  | 2.62 |
> > |        | H ∩ (S+G+Q)      | 41.86 | 28.29 | 8.79  | 0.00  | 0.00 |
> > | Mistral-7B-Instruct-v0.2 | S+G | 73.9 | 74.3 | 24.3 | 78.6 | 25.0 |
> > |        | S+Q              | 77.7  | 86.0  | 36.4 | 92.0  | 16.0 |
> > |        | S+G+Q            | 56.0  | 43.6  | 15.1 | 37.3  | 13.6 |
> > |        | H ∩ (S+G+Q)      | 65.3  | 40.8  | 14.3 | 47.7  | 5.8 |
> > | Yi-1.5-6B-Chat | S+G      | 81.67 | 63.82 | 23.68 | 85.71 | 14.29 |
> > |        | S+Q              | 54.33 | 56.40 | 21.20 | 38.00 | 14.00 |
> > |        | S+G+Q            | 60.37 | 41.02 | 15.49 | 30.97 | 11.02 |
> > |        | H ∩ (S+G+Q)      | 73.37 | 36.82 | 14.60 | 53.49 | 6.98 |
> >
> > ### **Toolkit Scope**
> >
> > We agree the toolkit should reflect broader geospatial workflows. In the revision (**Section 4.1(Page 6)**), we added an optical change-detection tool, alongside existing tools. The typed, hierarchical design is extensible, allowing easy integration of code-based and additional temporal operators. This ensures applicability to real-world pipelines while keeping the benchmark tractable.

---

### Official Review · Reviewer_e2yx · 2025-10-31

**Soundness:** 3
**Presentation:** 2
**Contribution:** 3
**Rating:** 4
**Confidence:** 4

**Summary:**

This paper introduces a new benchmark: GeoHOP, for remote-sensing-based geometrical reasoning tasks. The authors also proposed a baseline approach: GeoPlanner, combining LLMs with tool use. The authors have benchmarked several LLMs in the new benchmark with different settings.

**Strengths:**

- The benchmark is useful and fills in one gap compared to existing reasoning benchmarks for LLM (which are mostly indoor / not remote-sensing oriented).
- The paper is well written.
- The authors have benchmarked a wide range of LLMs from different companies.

**Weaknesses:**

**Experimental Evaluation Can be Stronger**

The analysis in the experiments is not super extensive/impressive. I would suggest the following as a benchmark paper:
- What are the failure modes that are common across all LLMs?
The authors have shown the Tool execution success/failures, but more importantly, to solve the problems in the benchmark, what is the current most challenging open problem? Currently, the paper has only a few such kinds of discussions (open problems and future directions) but this is very critical for a benchmark paper.
- What (fine-grained) capabilities can this benchmark evaluate?
Right now, it is a bit abstract that GeoHOP can evaluate "Geometric Reasoning in RS images", we would like some discussions on fine-grained difficulties that can be quantified. For example, the benchmark can have a subset of tasks where semantic perception/grounding is especially hard, a subset of tasks where geometric grounding / relational grounding is especially hard, and a subset of tasks where multi-step long-horizon reasoning/planning is especially hard, so that readers can develop methods to address each of these challenges.

**Related Works should be more extensive**

Though this paper is mainly focused on geospatial reasoning + perception, I encourage the authors to discuss a broader range of reasoning benchmarks for LLMs in related works. E.g., FOLIO [1] and LogiCity [2], they are also closely related literature that is trying to evaluate similar capabilities of LLMs (especially in LogiCity the visual track requires both perception/grounding and reasoning). The authors should highlight the key challenge in geospatial reasoning compared to the reasoning problems in those benchmarks.

[1] Han, Simeng, et al. "Folio: Natural language reasoning with first-order logic." arXiv preprint arXiv:2209.00840 (2022).
[2] Li, Bowen, et al. "LogiCity: Advancing neuro-symbolic ai with abstract urban simulation." Advances in Neural Information Processing Systems 37 (2024): 69840-69864.

**Questions:**

See weaknesses. I will consider raising the score if the authors address my concerns during the rebuttal phase.

---

> ### Author Response · Authors · 2025-11-26
> **Response to Weakness 1:**
>
> We thank the reviewer for the constructive feedback and helpful suggestions regarding experimental depth and related work. We address each point below.
> ### **Experimental Analysis and Failure Modes**
>
> In the revised manuscript (**Section 5.4**) (**Page 10**), we have added a unified error taxonomy covering:
>
>   (1) **Format errors** — invalid/incomplete JSON or argument fields.
>   (2) **Reasoning errors** — logical mistakes with correct tool/perception input.
>   (3) **Perception errors** — misinterpretation of visual/geospatial inputs, including masks, change maps, and spatial layouts.
>   (4) **Tool-use errors** — incorrect operator choice, missing steps, or invalid parameters (e.g., incorrect buffer distances).
>
> We also added **Table 4**, summarizing cross-model error distributions to highlight which failure modes are dominant across models. Results indicate that robust structured output generation and stable multimodal perceptual grounding—not abstract reasoning—are the key limitations for current geospatial agents, highlighting the need for better format alignment and stronger cross-modal grounding in future systems.
> ### **Table 4: Percentage distribution of four categories of errors for different LLMs**
>
> | **Model** | **Format Error (%)** | **Reasoning Error (%)** | **Perception Error (%)** | **Tool-Use Error (%)** |
> |-----------|----------------------|---------------------------|----------------------------|--------------------------|
> | **GPT-4o** | 41.65 | 13.49 | 41.26 | 3.60 |
> | **Gemini-2.5-Flash** | 29.47 | 21.18 | 12.09 | 37.26 |
> | **GPT-3.5** | 31.20 | 18.60 | 10.70 | 39.50 |
> | **DeepSeek-V3** | 66.40 | 23.00 | 8.70 | 1.90 |
> | **Qwen2.5-32B-Instruct** | 31.27 | 19.78 | 13.99 | 34.96 |
> | **Llama-3-70B-Instruct** | 55.20 | 25.70 | 18.20 | 0.90 |
> | **Qwen2.5-7B-Instruct** | 42.65 | 23.48 | 29.37 | 4.50 |
> | **InternLM3-8B-Instruct** | 32.70 | 23.90 | 11.40 | 32.00 |
> | **LLaMA3-1-8B** | 38.36 | 24.58 | 20.28 | 16.78 |
> | **Phi-3-Mini-4K-Instruct** | 45.60 | 34.70 | 1.80 | 17.90 |
> | **Mistral-7B-Instruct-v0.2** | 45.40 | 21.20 | 5.40 | 28.00 |
> | **Yi-1.5-6B-Chat** | 37.90 | 28.60 | 8.60 | 24.90 |
>
> ---
>
> ### **Related Work Coverage**
>
>
> In the revised **Related Work** (**Page 2**) section, we have added a dedicated subsection on reasoning benchmarks for LLMs beyond remote sensing, explicitly discussing **FOLIO (Han et al., 2022)** and **LogiCity (Li et al., 2024)**, among others. We emphasize that GeoHOP is complementary to these benchmarks: it inherits the structured, multi-step reasoning paradigm exemplified by FOLIO and LogiCity, but situates it within a geospatial context where
> (i) multi-modal perception,
> (ii) spatially grounded logic,
> (iii) specialized geospatial operations, and
> (iv) multi-scale contextual reasoning
> are essential to task success.

---

> ### Author Response · Authors · 2025-11-26
> **Response to Weakness 2:**
>
> We thank the reviewer for this insightful suggestion on fine-grained capability analysis.
> ### **Fine-Grained Capability Evaluation**
>
> To make GeoHOP’s evaluable capabilities explicit, we introduced capability tags for:
>
>   (1) **S**: Semantic perception.
>   (2) **G**: Geometric reasoning.
>   (3) **Q**: Quantitative reasoning.
>   (4) **H**: Long-horizon planning (≥4 tools).
>
> From these, we define four difficulty subsets:
>
>   (1) **S+G** — semantic + geometric reasoning tasks.
>   (2) **S+Q** — semantic + quantitative reasoning tasks.
>   (3) **S+G+Q** — full-pipeline tasks integrating all three.
>   (4) **H ∩ (S+G+Q)** — most challenging cases: long-horizon tasks with all three capability types.
>
> We present this in **Appendix A.1, Table 5** (**Page 16**) (“Fine-grained results across models and capability subsets”), reporting accuracy for instruction-following, tool selection, argument correctness, and summary generation, final answer. API-based models, especially GPT-4o, maintain strong performance across all subsets, particularly complex integrations like S+G+Q and long-horizon H ∩ (S+G+Q). Open-source models often perform well on simpler S+G and S+Q tasks, but suffer substantial degradation on multi-capability, long-horizon workflows—underscoring the difficulty of sustained, high-fidelity tool reasoning in extended geospatial pipelines.
> ### **Table 5 : Fine-grained results across models and capability subsets**
>
> | Model | Task | Inst. | Tool. | Arg. | Summ. | Ans. |
> |-------|------|-------|-------|------|-------|------|
> | **API-based** |  |  |  |  |  |  |
> | GPT-4o | S+G              | 77.78 | 92.11 | 32.89 | 75.00 | 45.29 |
> |        | S+Q              | 78.33 | 89.20 | 22.80 | 94.00 | 36.00 |
> |        | S+G+Q            | 85.20 | 84.40 | 27.37 | 91.08 | 35.43 |
> |        | H ∩ (S+G+Q)      | 60.70 | 46.77 | 18.09 | 60.47 | 9.81 |
> | Gemini-2.5-Flash | S+G    | 49.44 | 48.03 | 18.42 | 53.57 | 21.43 |
> |        | S+Q              | 30.67 | 25.60 | 12.00 | 30.00 | 8.00 |
> |        | S+G+Q            | 34.51 | 23.40 | 9.00  | 24.93 | 9.97 |
> |        | H ∩ (S+G+Q)      | 50.81 | 34.24 | 12.40 | 41.86 | 0.00 |
> | GPT-3.5 | S+G             | 67.2  | 73.0  | 27.0  | 64.3  | 42.9 |
> |        | S+Q              | 83.3  | 94.4  | 40.8  | 98.0  | 48.0 |
> |        | S+G+Q            | 53.9  | 53.5  | 20.8  | 51.2  | 24.7 |
> |        | H ∩ (S+G+Q)      | 67.1  | 54.4  | 18.3  | 69.8  | 10.5 |
> | **Open-source (Large)** |  |  |  |  |  |  |
> | DeepSeek-V3 | S+G         | 80.0  | 86.8  | 28.3  | 92.9  | 32.1 |
> |        | S+Q              | 83.3  | 96.8  | 30.0  | 98.0  | 40.0 |
> |        | S+G+Q            | 61.9  | 65.7  | 24.0  | 53.8  | 21.3 |
> |        | H ∩ (S+G+Q)      | 70.7  | 54.8  | 17.7  | 60.5  | 5.8 |
> | Qwen2.5-32B-Instruct | S+G | 59.4 | 65.1 | 23.7 | 60.7 | 10.7 |
> |        | S+Q              | 83.3  | 94.4  | 41.2 | 98.0  | 32.0 |
> |        | S+G+Q            | 60.1  | 57.5  | 15.4 | 52.0  | 21.3 |
> |        | H ∩ (S+G+Q)      | 66.0  | 46.5  | 15.2 | 66.3  | 5.8 |
> | Llama-3-70B-Instruct | S+G | 80.00 | 80.26 | 24.34 | 89.29 | 25.00 |
> |        | S+Q              | 83.33 | 92.80 | 38.80 | 94.00 | 34.00 |
> |        | S+G+Q            | 64.44 | 70.12 | 23.17 | 62.99 | 23.10 |
> |        | H ∩ (S+G+Q)      | 72.21 | 47.67 | 15.37 | 80.23 | 11.63 |
> | **Open-source (Medium/Small)** |  |  |  |  |  |  |
> | Qwen2.5-7B-Instruct | S+G  | 80.0  | 94.7  | 32.2  | 85.7  | 42.9 |
> |        | S+Q              | 83.3  | 96.0  | 32.4  | 100.0 | 34.0 |
> |        | S+G+Q            | 78.7  | 77.7  | 23.5  | 57.5  | 22.0 |
> |        | H ∩ (S+G+Q)      | 69.0  | 53.6  | 17.1  | 73.3  | 12.8 |
> | InternLM3-8B-Instruct | S+G | 74.4 | 82.9 | 28.9 | 78.6 | 25.0 |
> |        | S+Q              | 83.3  | 94.4  | 41.2 | 100.0 | 32.0 |
> |        | S+G+Q            | 55.8  | 55.2  | 22.2 | 38.6  | 19.7 |
> |        | H ∩ (S+G+Q)      | 70.2  | 49.6  | 16.8 | 64.0  | 5.8 |
> | LLaMA3-1-8B | S+G         | 80.0  | 89.5  | 28.3  | 85.7  | 21.4 |
> |        | S+Q              | 51.7  | 51.6  | 28.4 | 40.0  | 6.0 |
> |        | S+G+Q            | 71.2  | 71.0  | 23.2 | 50.7  | 19.2 |
> |        | H ∩ (S+G+Q)      | 67.3  | 48.7  | 16.0 | 76.7  | 10.5 |
> | Phi-3-Mini-4K-Instruct | S+G | 51.11 | 42.11 | 14.47 | 32.14 | 7.14 |
> |        | S+Q              | 77.67 | 82.00 | 32.40 | 72.00 | 16.00 |
> |        | S+G+Q            | 57.35 | 40.98 | 18.34 | 2.62  | 2.62 |
> |        | H ∩ (S+G+Q)      | 41.86 | 28.29 | 8.79  | 0.00  | 0.00 |
> | Mistral-7B-Instruct-v0.2 | S+G | 73.9 | 74.3 | 24.3 | 78.6 | 25.0 |
> |        | S+Q              | 77.7  | 86.0  | 36.4 | 92.0  | 16.0 |
> |        | S+G+Q            | 56.0  | 43.6  | 15.1 | 37.3  | 13.6 |
> |        | H ∩ (S+G+Q)      | 65.3  | 40.8  | 14.3 | 47.7  | 5.8 |
> | Yi-1.5-6B-Chat | S+G      | 81.67 | 63.82 | 23.68 | 85.71 | 14.29 |
> |        | S+Q              | 54.33 | 56.40 | 21.20 | 38.00 | 14.00 |
> |        | S+G+Q            | 60.37 | 41.02 | 15.49 | 30.97 | 11.02 |
> |        | H ∩ (S+G+Q)      | 73.37 | 36.82 | 14.60 | 53.49 | 6.98 |

---

> ### Comment · Reviewer_e2yx · 2025-11-27
> **Thanks for the rebuttal**
>
> I appreciate the authors' rebuttal, and my concerns have been addressed; therefore, I will change my evaluation to 6.

---

### Official Review · Reviewer_5bSx · 2025-11-01

**Soundness:** 3
**Presentation:** 3
**Contribution:** 2
**Rating:** 4
**Confidence:** 4

**Summary:**

This paper proposes GeoHOP, a benchmark dataset for evaluating the capabilities of LLM-driven tool augmentation agents on geospatial reasoning tasks. GeoHOP contains 417 scene-driven hierarchical perception tasks covering optical, SAR, and infrared imagery. The authors also propose GeoPlanner, an LLM-driven agent that organizes analytics tools into a functional hierarchy and executes a fault-tolerant inference pipeline. Experiments show that even state-of-the-art models still have significant room for improvement on geospatial cognitive reasoning tasks.

**Strengths:**

1. The paper makes a compelling case for moving beyond perception-oriented tasks to cognitive-level geospatial reasoning. The distinction between simply detecting objects versus understanding their spatial relationships and implications for real-world decisions is clearly articulated.

2. The two-stage pipeline with knowledge-augmented generation followed by expert adjudication demonstrates careful attention to quality. Having eight domain experts perform three-pass reviews with an auditable rubric shows commitment to data integrity.

**Weaknesses:**

1. With only 417 instances, the dataset is relatively small compared to other benchmarks in the field. The reliance on just four source datasets may not capture the full complexity of real-world geospatial scenarios, particularly multi-temporal analyses or extreme weather events that practitioners actually encounter.
2. Using LLM-as-judge for evaluation introduces potential biases that aren't adequately addressed. The lack of human validation or inter-rater reliability measures undermines confidence in the results. Additionally, the absence of fine-grained error analysis makes it difficult to understand failure modes.
3. GeoPlanner largely combines existing techniques (hierarchical planning, ReAct-style execution, standard sentence embeddings). The contribution feels more like engineering than research innovation, which may limit its impact at a top-tier venue.
4. The absence of ablation studies leaves key design choices unjustified. Without comparing to ThinkGeo directly or analyzing performance differences across modalities, it's hard to assess the true benefits of the proposed approach. The lack of computational cost analysis is also problematic for practical adoption.

**Questions:**

1. How is the quality of queries and toolchains generated by ChatGPT-5 ensured? How consistent is the expert review process?

2. What are the performance differences of GeoPlanner across different modalities (optical vs. SAR vs. IR)? Are modality-specific adjustments required?

3. What are the most common failure modes? Is it due to incorrect tool selection or parameter settings?

---

> ### Author Response · Authors · 2025-11-26
> **Response to Weakness:**
>
> We thank the reviewer for the thoughtful and constructive feedback.
> We appreciate the careful assessment of our work and address each point below.
> ### **Dataset Scale & Coverage**
>
> In the revised manuscript, **Section 3.2, Table 1 (Page 5)** expands the benchmark from **417 to 545** instances by adding **3 new datasets**, including **2 city diversified land cover datasets** and a semantic **change-detection dataset**, which directly increases **multi-temporal** and scene diversity.
>
> We also add **Appendix A.6 (Scene-Context Knowledge Base) (Page 20)**, which lists all domain-grounded scene types and example queries, demonstrating that GeoHOP captures substantially richer spatial complexity. This includes varied land-cover categories, spatial configurations, and operational contexts relevant to real-world geospatial reasoning.
> ### **Table 1: Datasets used as image sources in the GeoHOP benchmark**
>
> | **Name** | **Annotation Type** | **Resolution** | **Modality** | **Geographical Coverage** |
> |---------|----------------------|----------------|--------------|----------------------------|
> | LoveDA  | Masks | 0.3 m/px | Optical | **Nanjing / Changzhou / Wuhan, China** |
> | ISPRS Potsdam | Masks | 0.05 m/px | Optical | **Potsdam, Germany** |
> | **ISPRS Vaihingen** | Masks | 0.09 m/px | Optical | **Vaihingen, Germany** |
> | **DeepGlobe Land Cover** | Masks | 0.50 m/px | Optical | **Cities in India / Indonesia / Thailand** |
> | **HRSCD** | Change masks | 0.45 m/px | Optical | **Rennes / Caen, France** |
> | OGSOD | Bounding boxes | 3 m/px | SAR | **Cities in China** |
> | DMIST | Bounding boxes | -- | IR | **Chengdu, China** |
>
> ---
> ### **Evaluation Method: LLM-as-Judge Bias:**
>
> While programmatic evaluation is ideal, many GeoHOP tasks require **joint assessment of numeric values and scene-based explanations**, which—as illustrated in the new example (**Figure 4**) (**Page 9**)—cannot be scored by deterministic matching alone. In such cases, an LLM-as-judge is necessary.
>
> To address reliability concerns: (1) We added a fine-grained error analysis (**Section 5.4, Table 4**) (**Page 10**), highlighting common error types. (2) We employ structured, reference-grounded prompts to guide LLM evaluation and reduce bias. This combination improves transparency and minimizes evaluation subjectivity.
> ### **Table 4: Percentage distribution of four categories of errors for different LLMs**
>
> | **Model** | **Format Error (%)** | **Reasoning Error (%)** | **Perception Error (%)** | **Tool-Use Error (%)** |
> |-----------|----------------------|---------------------------|----------------------------|--------------------------|
> | **GPT-4o** | 41.65 | 13.49 | 41.26 | 3.60 |
> | **Gemini-2.5-Flash** | 29.47 | 21.18 | 12.09 | 37.26 |
> | **GPT-3.5** | 31.20 | 18.60 | 10.70 | 39.50 |
> | **DeepSeek-V3** | 66.40 | 23.00 | 8.70 | 1.90 |
> | **Qwen2.5-32B-Instruct** | 31.27 | 19.78 | 13.99 | 34.96 |
> | **Llama-3-70B-Instruct** | 55.20 | 25.70 | 18.20 | 0.90 |
> | **Qwen2.5-7B-Instruct** | 42.65 | 23.48 | 29.37 | 4.50 |
> | **InternLM3-8B-Instruct** | 32.70 | 23.90 | 11.40 | 32.00 |
> | **LLaMA3-1-8B** | 38.36 | 24.58 | 20.28 | 16.78 |
> | **Phi-3-Mini-4K-Instruct** | 45.60 | 34.70 | 1.80 | 17.90 |
> | **Mistral-7B-Instruct-v0.2** | 45.40 | 21.20 | 5.40 | 28.00 |
> | **Yi-1.5-6B-Chat** | 37.90 | 28.60 | 8.60 | 24.90 |
>
> ---
> ### **Novelty & Contribution:**
>
> Our contribution lies in pioneering an underexplored domain: geospatial cognitive reasoning. GeoHOP provides **545** expert-curated, high-complexity instances explicitly grounded in spatial logic.
>
> As shown in **Appendices A.3 (Page 18), A.5 (Page 19), and A.6 (Page 20)**:
>
>   (1) Existing RS-MLLMs fail to generalize to GeoHOP tasks (**Table 7**).
>   (2) No prior benchmark offers comparable multi-tool, multi-modal reasoning scenarios (**Figure 8**).
>   (3) Each instance requires substantial domain semantics and integration of perception, geometry, and quantitative reasoning (**Figure 9–12**).
>
> We believe these aspects reflect substantive novelty beyond engineering integration.
> ### **Table 7: Optical accuracy comparison between representative remote-sensing multimodal LLMs (RS-MLLMs) and our GeoPlanner (GPT-4o) on GeoHOP optical VQA tasks.**
>
>
> | **Model** | **Optical Accuracy (%)** |
> |----------|----------------------------|
> | **RS-MLLMs** | |
> | EarthDial [1]| 9.38 |
> | GeoChat [2] | 6.70 |
> | GeoPix [3] | 6.17 |
> | **LLM-based Agent (Ours)** | |
> | **GeoPlanner (GPT-4o)** | **29.29** |
>
> References:
> [1] Soni, Sagar, et al. "EarthDial: Turning multi-sensory earth observations to interactive dialogues." CVPR (2025).
> [2] Kuckreja, Kartik, et al. "GeoChat: Grounded large vision-language model for remote sensing." CVPR (2024).
> [3] Ou, Ruizhe, et al. "GeoPix: A multimodal large language model for pixel-level image understanding in remote sensing." IEEE GRSM (2025).

---

> ### Author Response · Authors · 2025-11-26
> **Response to Questions:**
>
> We thank the reviewer for raising these important questions. We address them point-by-point below.
> ### **Quality control of LLM-generated queries and expert adjudication**
>
>
> We ensure query/tool-chain quality through a two-stage pipeline (**Section 3.2**) (**Page 3**):
>
>   (1) Instances are generated under explicit hardness controls and operator-legality checks.
>   (2) Each undergoes a three-pass hierarchical expert review assessing semantic integrity, tool-sequence coherence, and geometric correctness.
>
> Backend API logs show each fully validated instance costs approximately **$1.4** to generate, reflecting the significant expert effort invested in curation and quality assurance.
>
> ---
>
> ### **Modality-specific performance differences**
>
> The revised **Appendix A.2 (Table 6)** (**Page 17**) now reports modality-dependent accuracy, showing expected differences across Optical/SAR/IR:
>
>   (1) **Optical:** moderate accuracy for high-capacity controllers.
>   (2) **SAR:** largest drop due to texture/speckle noise; many open-source models perform poorly.
>   (3) **IR:** frontier models achieve near-perfect scores, smaller models remain below 45%.
>
> GeoPlanner requires no modality-specific tuning: tool retrieval is semantic and modality-tagged, with automatic selection of compatible operators.
> ### **Table 6: Accuracy comparison across models on the overall benchmark and on each sensing modality (Optical, SAR, IR).**
>
> | **Model** | **Overall** | **Optical** | **SAR** | **IR** |
> |-----------|-------------|-------------|---------|--------|
> | **GPT-4o** | **25.70** | **29.29** | 11.50 | **100.00** |
> | GPT-3.5 | 24.40 | 28.05 | 0.00 | 56.82 |
> | Gemini-2.5-Flash | 12.04 | 11.63 | 0.00 | **100.00** |
> | DeepSeek-V3 | 21.28 | 25.19 | 0.00 | 43.18 |
> | Qwen2.5-32B-Instruct | 19.08 | 23.12 | 0.00 | 34.09 |
> | Qwen2.5-7B-Instruct | 22.57 | 22.08 | 17.24 | 40.91 |
> | Llama-3.1-8B-Instruct | 16.70 | 14.81 | **26.72** | 6.82 |
> | Llama-3-70B-Instruct | 22.75 | 27.27 | 0.00 | 43.18 |
> | InternLM3-8B-Instruct | 18.90 | 22.08 | 0.86 | 38.64 |
> | Mistral-7B-Instruct-v0.2 | 13.21 | 16.36 | 0.00 | 20.45 |
> | Phi-3-Mini-4K-Instruct | 3.67 | 0.52 | 8.62 | 18.18 |
> | Yi-1.5-6B-Chat | 10.83 | 13.51 | 0.00 | 15.91 |
>
> ---
>
> ### **Failure Mode Analysis**
>
> In the revised version, we also added a fine-grained error breakdown (**Section 5.4, Table 4**) (**Page 10**) showing that the most frequent failures are format errors and perception errors, not abstract reasoning mistakes.
> ### **Table 10 : Percentage distribution of four categories of errors for different LLMs**
>
> | **Model** | **Format Error (%)** | **Reasoning Error (%)** | **Perception Error (%)** | **Tool-Use Error (%)** |
> |-----------|----------------------|---------------------------|----------------------------|--------------------------|
> | **GPT-4o** | 41.65 | 13.49 | 41.26 | 3.60 |
> | **Gemini-2.5-Flash** | 29.47 | 21.18 | 12.09 | 37.26 |
> | **GPT-3.5** | 31.20 | 18.60 | 10.70 | 39.50 |
> | **DeepSeek-V3** | 66.40 | 23.00 | 8.70 | 1.90 |
> | **Qwen2.5-32B-Instruct** | 31.27 | 19.78 | 13.99 | 34.96 |
> | **Llama-3-70B-Instruct** | 55.20 | 25.70 | 18.20 | 0.90 |
> | **Qwen2.5-7B-Instruct** | 42.65 | 23.48 | 29.37 | 4.50 |
> | **InternLM3-8B-Instruct** | 32.70 | 23.90 | 11.40 | 32.00 |
> | **LLaMA3-1-8B** | 38.36 | 24.58 | 20.28 | 16.78 |
> | **Phi-3-Mini-4K-Instruct** | 45.60 | 34.70 | 1.80 | 17.90 |
> | **Mistral-7B-Instruct-v0.2** | 45.40 | 21.20 | 5.40 | 28.00 |
> | **Yi-1.5-6B-Chat** | 37.90 | 28.60 | 8.60 | 24.90 |

---

> ### Author Response · Authors · 2025-12-03
> **Response to Weakness2:**
>
> We thank the reviewer for pointing out the necessity of ablation studies.
>
> **Ablation Studies**
>
> We have added a new section **(Appendix A.8, Table 9)(Page 26)** comparing GeoPlanner with a standard ReAct baseline. These findings empirically justify that our structured architecture is essential for robust geospatial reasoning.
>
> **Table 9**: Ablation study comparing the full GeoPlanner framework against ThinkGeo (a Standard ReAct baseline) using GPT-4o. Inst., Tool., Arg., and Summ. denote step-by-step accuracies, while P, O, L, and Ans. represent end-to-end execution and reasoning scores.
>
>
> | Method | Inst. | Tool. | Arg. | Summ. | P | O | L | Ans. |
> | :--- | :---: | :---: | :---: | :---: | :---: | :---: | :---: | :---: |
> | ThinkGeo [1] (ReAct) | 77.46 | 48.11 | 14.18 | 46.97 | 48.05 | 5.98 | 50.87 | 8.62 |
> | **GeoPlanner** | **79.40** | **72.01** | **23.57** | **82.83** | **72.73** | **88.38** | **77.95** | **25.70** |
>
> Reference:
> [1]:Shabbir, A. et al. (2025). ThinkGeo: Evaluating Tool-Augmented Agents for Remote Sensing Tasks. arXiv:2505.23752.

---

### Author Response · Authors · 2025-12-02
**Official Comment by Authors**

**Dear Program Chairs, Senior Area Chairs, Area Chairs, and Reviewers,**

We sincerely thank you for your constructive feedback and expert insights. We are encouraged by the positive engagement during the rebuttal phase and the validation of **GeoHOP** as a critical contribution moving beyond perception to expert-verified geospatial reasoning.

**Update on Rebuttal Progress:**
We are pleased to report substantial progress following our response and manuscript revisions. Specifically, **Reviewer e2yx has confirmed that his/her concerns were fully addressed and has raised his/her evaluation score to 6**. **Reviewer CA8L**, who explicitly acknowledged the **quality improvement** and **signaled an intention to raise his/her score to 6**. We include the reviewer's official comment for clarity:

---

**Reviewer e2yx:** "**I appreciate the authors' rebuttal, and my concerns have been addressed; therefore, I will change my evaluation to 6.**" (Official Comment)

---

**Reviewer CA8L:** "**We sincerely thank the authors for their tremendous effort; the quality of the manuscript has indeed improved substantially. This is an important and valuable line of work.**" (Official Comment)

---

While we await further replies from **Reviewers 5bSx, PTWu, and guxm**, we have proactively integrated **comprehensive solutions** for all their raised concerns (e.g., detailed error analysis, paradigm comparisons) into the revised manuscript to ensure a **thorough and robust improvement of the submission**.

Based on this valuable feedback, we have implemented extensive revisions and added new experiments. The key updates are summarized below:

1.  **Benchmark Expansion (Section 3.2, Table 1, Page 5):** We expanded GeoHOP from 417 to **545 instances** by incorporating three new datasets (including diverse city land-cover and semantic change detection). This directly enhances multi-temporal and scene diversity. A "Geographical Coverage" column was added to Table 1 to illustrate data distribution.
2.  **New Tool Integration (Section 4.1, Page 6):** We integrated a **Change Detection** tool into the GeoPlanner agent, significantly enhancing its capability to handle complex dynamic reasoning tasks.
3.  **Expanded Baselines & Metrics (Section 5.2, Table 3, Page 8):** We evaluated additional state-of-the-art LLMs on GeoHOP and introduced three specific evaluation metrics—P (Perception), O (Spatial Statistics), and L(Spatial Relations)—for a more rigorous assessment.
4.  **Error Analysis (Section 5.4, Table 4, Page 10):** We added a comprehensive error analysis to categorize failure modes across different models, offering transparency into current limitations.
5.  **Fine-Grained Difficulty Analysis (Appendix A.1, Table 5, Page 16):** We dissected model performance across varying levels of task complexity .
6.  **Modality-Specific Evaluation (Appendix A.2, Table 6, Page 17):** We provided a breakdown of accuracy across Optical, SAR, and IR modalities, highlighting significant challenges in non-optical domains.
7.  **Comparison with RS-MLLMs (Appendix A.3, Table 7, Page 18):** To benchmark against existing paradigms, we added a direct comparison between our GeoPlanner (GPT-4o) and representative Remote Sensing Multimodal LLMs (RS-MLLMs) on optical VQA tasks.
8.  **Computational Cost Analysis (Appendix A.4, Figures 6 & 7, Page 19):** We provided detailed metrics on per-task inference latency, total end-to-end runtime, and a breakdown of tool usage (LLM reasoning vs. perception vs. spatial analysis).
9.  **Benchmark Comparison Overview (Appendix A.5, Figure 8, Page 20):** We added a visual comparison placing GeoHOP examples alongside those from existing RS benchmarks, directly highlighting the difference in scope and reasoning depth.
10. **Scene Context & Domain Knowledge (Appendix A.6, Page 20):** We included domain-knowledge-grounded scene-context examples (e.g., industrial safety zones, urban heat islands) to demonstrate tasks requiring contextual interpretation.
11. **Agent Paradigm Comparison (Appendix A.7, Table 8, Page 25):** We added an experimental comparison between our **Tool-Augmented Agent** and a **Code-Based Agent** (Cursor). This section discusses the trade-offs between reasoning flexibility and execution stability, addressing specific suggestions regarding future hybrid designs.
12. **Ablation Studies (Appendix A.8, Table 9, Page 26)**: We added an ablation study comparing GeoPlanner against ThinkGeo (a standard ReAct baseline).The results demonstrate that our Hierarchical Tool Organization and Error-Aware Replanning mechanisms significantly enhance tool retrieval accuracy and execution stability

We have incorporated all the above additions into the revised manuscript. We believe these supplementary experiments and analyses effectively address the reviewers' concerns and solidify the paper's contribution.

Thank you again for your time and dedication to improving our work.

Sincerely,

**The Authors of Submission #25567**

---

### Meta-Review · Area_Chair_fAo6 · 2026-01-07

**Summary:**

This paper presents a benchmark for hierarchical geospatial reasoning, and GeoPlanner, a tool-augmented LLM agent evaluated on that benchmark.

Following reviewer feedback, the authors substantially revised the manuscript: they expanded the dataset (from 417 to 545 instances), added modality and fine-grained analyses, introduced an error taxonomy and ablation studies, added a change-detection tool and bi-temporal tasks, provided computational-cost measurements, and ran a small exploratory comparison to a code-based agent (Cursor). Two reviewers expressed increased confidence after the rebuttal.

Despite these improvements, I recommend rejection at this time because key methodological and reporting weaknesses remain. The empirical claims and the reliability of the benchmark are not yet fully validated.

**Reviewer Concerns:**

Major concerns are:

1. Evaluation validity remains unproven (LLM-as-judge not validated).
2. Human verification / dataset quality statistics are missing.
3. The size of the dataset is small
4. Reproducibility and release plan incomplete.

These are outstanding concerns.

**Reviewer Scores:**

Two may have increased their scores, two may not, however some of the concerns are still open.

---

### Decision · Program_Chairs · 2026-01-26

Reject